# Chromosome alignment maintenance requires the MAP RECQL4, mutated in the Rothmund–Thomson syndrome

Hideki Yokoyama[1,2,3,]*, Daniel Moreno-Andres[1,2,]*, Susanne A Astrinidis[1], Yuqing Hao[4], Marion Weberruss[1,2], Anna K Schellhaus[1,2], Hongqi Lue[2], Yoshikazu Haramoto[5], Oliver J Gruss[6], Wolfram Antonin[1,2]

RecQ-like helicase 4 (RECQL4) is mutated in patients suffering from the Rothmund–Thomson syndrome, a genetic disease characterized by premature aging, skeletal malformations, and high cancer susceptibility. Known roles of RECQL4 in DNA replication and repair provide a possible explanation of chromosome instability observed in patient cells. Here, we demonstrate that RECQL4 is a microtubule-associated protein (MAP) localizing to the mitotic spindle. RECQL4 depletion in M-phase–arrested frog egg extracts does not affect spindle assembly per se, but interferes with maintaining chromosome alignment at the metaphase plate. Low doses of nocodazole depolymerize RECQL4-depleted spindles more easily, suggesting abnormal microtubule–kinetochore interaction. Surprisingly, inter-kinetochore distance of sister chromatids is larger in depleted extracts and patient fibroblasts. Consistent with a role to maintain stable chromosome alignment, RECQL4 down-regulation in HeLa cells causes chromosome misalignment and delays mitotic progression. Importantly, these chromosome alignment defects are independent from RECQL4's reported roles in DNA replication and damage repair. Our data elucidate a novel function of RECQL4 in mitosis, and defects in mitotic chromosome alignment might be a contributing factor for the Rothmund–Thomson syndrome.

## Introduction

Mutations in RECQL4, one of the five helicases of the RECQ family in humans, cause the Rothmund–Thomson syndrome, a rare autosomal recessive disease. The disease is defined by chromosome fragility; premature aging characterized by rash skin, hair loss, and cataracts; developmental abnormalities such as skeletal malformationsl and predisposition for cancer, particularly osteosarcoma (Kitao et al, 1999; Croteau et al, 2012b). Distinct RECQL4 mutations are also linked to the RAPADILINO syndrome, indicated by skeletal malformations but no cancer predisposition (Siitonen et al, 2003), and the Baller–Gerold syndrome, characterized by bone abnormalities of the skull, arms, and hands (Van Maldergem et al, 2006). A gene deletion of RECQL4 in mice is lethal in early development (Ichikawa et al, 2002). A hypomorphic mutation deleting a single exon leads to growth retardation and developmental abnormalities (Hoki et al, 2003), whereas exon deletions causing truncation of the C-terminal part of RECQL4 result in aneuploidy and cancer predisposition in mice (Mann et al, 2005).

On a molecular level, RECQL4 shows weak DNA helicase activity in vitro (Xu & Liu, 2009) and is involved in DNA replication (Sangrithi et al, 2005; Matsuno et al, 2006), DNA damage response (Kumata et al, 2007; Lu et al, 2016), and telomere maintenance (Ghosh et al, 2012). RECQL4 function in DNA replication requires its N-terminal domain, which resembles the Saccharomyces cerevisiae Sld2p protein (Matsuno et al, 2006) but is not affected by disease-causing mutations (Siitonen et al, 2009). Consistent with the above functions, RECQL4 localizes to the nucleus (Yin et al, 2004; Petkovic et al, 2005; Woo et al, 2006) but also to the mitochondria (Singh et al, 2010; Croteau et al, 2012a) where it is involved in maintaining mitochondrial DNA integrity. Thus, RECQL4 participates in a variety of cellular processes. Yet, it is unresolved which primary functions of RECQL4 are defective in the different diseases and, hence, the loss of which function is causative for the described pathological phenotypes.

We have previously described potential mitosis-specific microtubule-associated proteins (MAPs) identified by a sequential microtubule and import receptor binding (Yokoyama et al, 2009, 2013, 2014). The same pull-down strategy identified RECQL4 as a potential MAP (data not shown), a finding which we further investigate here. Many nuclear proteins act in mitosis as microtubule regulators and enable spindle assembly (Cavazza & Vernos, 2015; Yokoyama, 2016). These MAPs generally possess a NLS targeting them to the nucleus in interphase. Accordingly, during this phase of the cell cycle they do not interact with and, thus, cannot regulate microtubules located in the cytoplasm. Upon mitotic nuclear

[1]Friedrich Miescher Laboratory of the Max Planck Society, Tübingen, Germany   [2]Institute of Biochemistry and Molecular Cell Biology, Medical School, Rheinisch-Westfälische Technische Hochschule Aachen University, Aachen, Germany   [3]ID Pharma Co. Ltd., Tsukuba, Japan   [4]Zentrum für Molekulare Biologie der Universität Heidelberg (ZMBH), Deutsches Krebsforschungszentrum-ZMBH Alliance, Heidelberg, Germany   [5]Biotechnology Research Institute for Drug Discovery, National Institute of Advanced Industrial Science and Technology, Tsukuba, Japan   [6]Institute of Genetics, Rheinische Friedrich-Wilhelms Universität Bonn, Bonn, Germany

Correspondence: hideki-yokoyama@idpharma.jp; wantonin@ukaachen.de
*Hideki Yokoyama and Daniel Moreno-Andres contributed equally to this work.

envelope breakdown, these MAPs get access to microtubules and regulate microtubule behavior locally around chromatin. The GTP-bound form of the small GTPase Ran (RanGTP), generated around chromatin, binds to nuclear transport receptors such as importin $\beta$, liberating the NLS-containing nuclear MAPs from the receptors. Each Ran-regulated MAP identified so far plays a distinct role in microtubule regulation to assemble a bipolar spindle. For example, TPX2 (targeting protein for Xklp2) promotes de novo microtubule nucleation around chromatin (Gruss et al, 2001), whereas CHD4 (chromodomain helicase DNA–binding protein 4) stabilizes and elongates already existing microtubules (Yokoyama et al, 2013), and kinesin-14 motor bundles the elongated microtubules (Weaver et al, 2015).

Here, we show that RECQL4 is a so far unrecognized MAP that localizes to spindle microtubules. RECQL4 is not required for spindle assembly per se, but is important for stable chromosome alignment to the metaphase plate.

# Results

## RECQL4 is a microtubule-associated protein

We identified RECQL4 as an NLS-containing potential MAP by a previously established (Yokoyama et al, 2013) sequential purification strategy of microtubule and importin-$\beta$-binding proteins (data not shown). To test whether RECQL4 can indeed interact with microtubules, we added taxol-stabilized microtubules to HeLa nuclear extracts containing RECQL4. Endogenous RECQL4 was efficiently co-sedimented with microtubules, indicating microtubule binding (Figs 1A and S1A) as detected with an antibody against human RECQL4 (Fig S1B). Addition of recombinant importin $\alpha/\beta$ complex prevented RECQL4–microtubule interaction (Fig 1A), as seen before for the MAPs imitation SWI and CHD4 (Yokoyama et al, 2009, 2013). Inhibition was reverted by the co-addition of RanGTP, which binds to importin $\beta$ and removes the importin complex from NLS sites. As previously reported, the microtubule polymerase chTOG, the orthologue of *Xenopus* XMAP215, showed no regulation by importins nor Ran (Yokoyama et al, 2014). Endogenous RECQL4 could also be co-sedimented from *Xenopus* egg extracts with taxol-stabilized microtubules (Fig S1C).

To test whether RECQL4 can directly interact with microtubules, we used recombinant *Xenopus* RECQL4, produced in insect cells (Fig S1D). Recombinant RECQL4 was co-pelleted with pure taxol-stabilized microtubules, indicating direct microtubule binding (in two independent experiments, 100% of the protein was detected in the pellet when co-sedimenting with MT versus 0% and 6.9% in the absence of MT), whereas contaminating proteins in the fraction did not (Fig 1B). Similar to what was observed in HeLa nuclear extracts, microtubule interaction of recombinant RECQL4 was blocked by addition of import $\alpha/\beta$ complex in a RanGTP-sensitive manner (Fig S1E).

## RECQL4 down-regulation in HeLa cells causes spindle defects

These data indicate that RECQL4 is indeed a Ran-regulated MAP. To assess its impact on microtubule function in cells, we analyzed HeLa cells stably expressing histone H2B-mCherry and EGFP-

$\alpha$-tubulin during mitotic progression. RECQL4 expression was efficiently down-regulated with each of three siRNA oligos (Fig S2A). 24 h post-transfection, live-cell imaging was carried out for 48 h (Fig 1C). Upon RECQL4 down-regulation, misaligned chromosomes (indicated by arrows) were detected in 20–25% of tracks in RECQL4–down-regulated cells as compared with 11% in controls, whereas the shape and size of the mitotic spindle were unchanged compared with control-treated cells (Fig 1D). Although we observed efficient RECQL4 down-regulation with each of the three siRNA oligos 48 and 72 h post-transfection (Fig S2A), we cannot exclude that the differences between the three oligos arise from slightly diverse depletion efficiencies. Systematic analysis of chromatin and microtubule features using the CellCognition software (Held et al, 2010) showed that the time from prophase to the anaphase chromosome segregation was significantly extended upon RECQL4 down-regulation (Fig 1E).

The number of lagging chromosomes and chromosome bridges were not significantly increased upon RECQL4 down-regulation. In immunofluorescence experiments, 4% of control cells in late anaphase show lagging chromosomes, whereas this percentage ranged from 4 to 23% in RECQL4–down-regulated cells using the three oligos. In the same experiments, 17% of control cells show chromosome bridges in late anaphase, whereas 7 to 24% of the RECQL4 down-regulated cells. Similarly, the number of ultra-fine chromatin bridges in anaphase, detected by PICH staining (Chan & Hickson, 2011), did not significantly change upon RECQL4 down-regulation (seen in 45% of control cells and between 23 and 56% in the RECQL4–down-regulated cells with the three oligos).

Next, we tested whether imbalance in mitotic microtubule dynamics could be responsible for chromosome misalignments. We used a recently developed assay (Stolz et al, 2015) where inhibition of the mitotic kinesin Eg5 by monastrol prevents centrosome separation at the beginning of mitosis. This causes circular symmetric monoasters in control cells, as observed by an $\alpha$-tubulin staining. Down-regulation of microtubule regulators including the plus end–stabilizing factors CLIP-170, CLASPs (Stolz et al, 2015), the microtubule bundling protein DRG1 (Schellhaus et al, 2017), or the centrosome proteins NuMA and PCM1 (Stolz et al, 2015) generates asymmetric monoasters under these conditions when compared with the control. Asymmetric monoasters show a characteristic triangular distribution of the $\alpha$-tubulin staining with the main density not locating in the center of the chromatin mass and of the CREST staining (Schellhaus et al, 2017; Stolz et al, 2015). Indeed, spindles in cells with reduced RECQL4 levels showed many more asymmetric asters (Fig 1F and G). This phenotype was rescued by addition of low doses of taxol, similar to what has been observed for other microtubule regulators such as CLASP1 and DRG1 (Stolz et al, 2015; Schellhaus et al, 2017). Thus, down-regulation of RECQL4 expression in HeLa cells causes spindle microtubule defects supporting the idea that RECQL4 has an important function as a mitotic MAP.

## Fibroblasts from Rothmund–Thomson syndrome patients show spindle abnormalities

Mutations in RECQL4 cause the Rothmund–Thomson syndrome, characterized by premature aging and susceptibility to certain

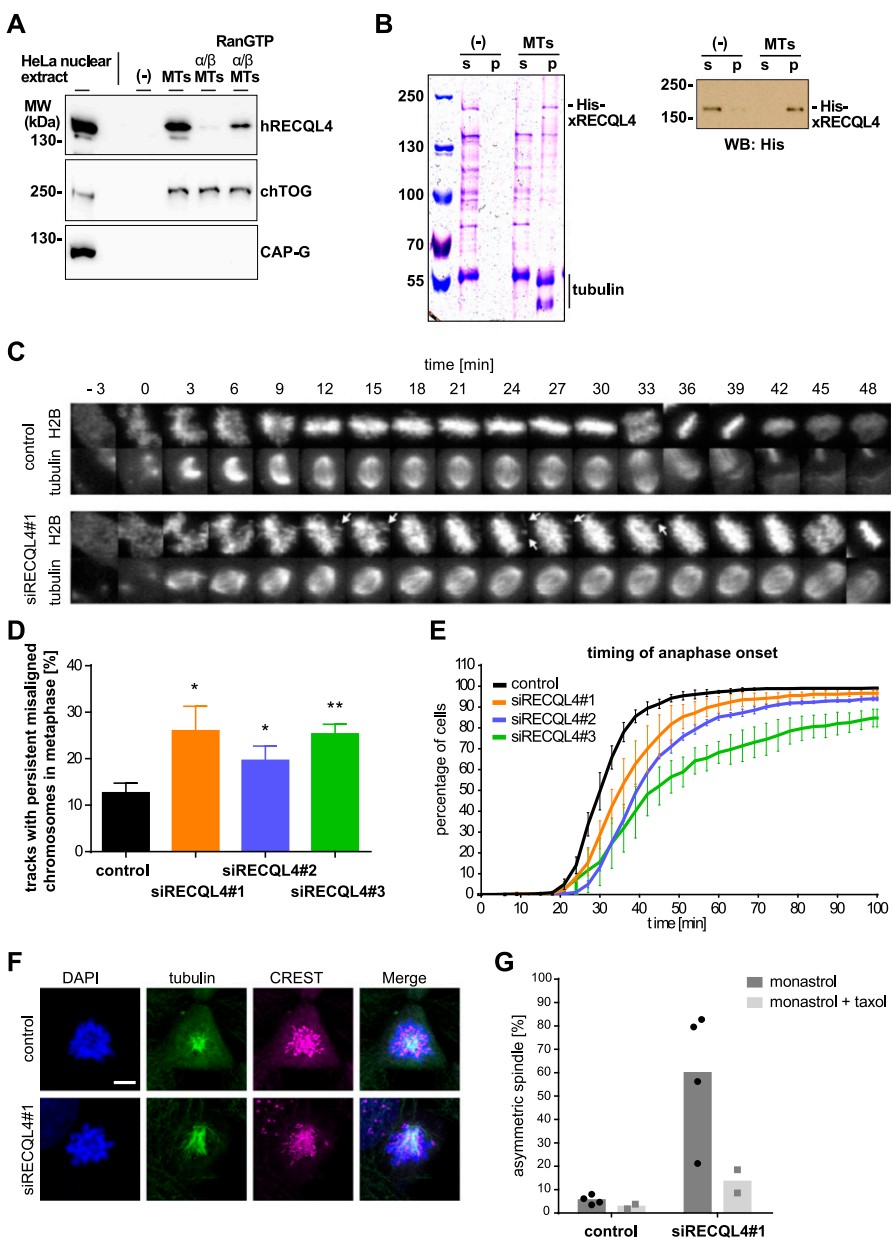

**Figure 1. RECQL4 is a MAP with a spindle function.**
**(A)** Human RECQL4 binds to microtubules (MTs) in a RanGTP-regulated manner. HeLa nuclear extract (1 mg/ml) was incubated with 2 $\mu$M pure taxol-stabilized MTs, in the presence or absence of recombinant importin $\alpha/\beta$ complex and RanGTP, and pelleted. MAPs were eluted with high salt from the pellet, and the supernatant after a second centrifugation was analyzed by immunoblot. **(B)** RECQL4 directly binds to MTs. 0.1 $\mu$M recombinant RECQL4 was incubated in the absence or presence of 2 $\mu$M taxol-stabilized MTs. Samples were separated by centrifugation and the supernatant (s) and pellet (p) fractions were analyzed by Coomassie staining and Western blot (WB) against His$_6$-tag. **(C)** HeLa cells, stably expressing mCherry-H2B and EGFP-$\alpha$-tubulin, were imaged for 48 h starting at 24 h post-transfection in intervals of 3 min. A representative track through mitosis is shown from control transfected and RECQL4–down-regulated cells. White arrows show misaligned chromosomes. **(D)** Quantification of chromosome misalignment in metaphase. Persistent misaligned chromosomes, as shown in (C), were quantitated in more than 100 cell tracks per siRNA in each of the three independent experiments. Error bars: SD. **$P < 0.01$; *$P < 0.05$ ($t$ test, two-tailed). **(E)** RECQL4 down-regulation slows down mitotic progression. Timing from prophase (0 min) to anaphase onset based on chromatin morphology is shown for the cells treated with control and three different RECQL4 siRNAs. Using data from more than 100 mitotic cell tracks per experiment, three independent experiments are plotted. Error bars: SD. **(F)** Representative immunofluorescence images from HeLa cells transfected for 72 h with RECQL4 siRNA showing asymmetric monopolar spindles, or control siRNA showing symmetric monopolar spindles. Cells were incubated with 70 $\mu$M of the kinesin-5/Eg5 inhibitor monastrol, fixed, and stained with DAPI (blue) and antibodies against $\alpha$-tubulin (green) and human centromere (CREST, magenta). Scale bar 5 $\mu$m. **(G)** Quantitation shows the percentage of asymmetric monopolar spindles after monastrol treatment in the absence (four independent experiments) or presence of taxol (two independent experiments). More than 22 cells with monopolar spindles were evaluated per data points. Black dots and grey squares indicate the mean of each independent experiment. *$P < 0.05$ ($t$ test, two-tailed).

cancers (Croteau et al, 2012b). To test whether spindle defects might be linked to the pathology, we analyzed two fibroblast cell lines (AG05013 and AG18371) from Rothmund–Thomson syndrome patients, which carry mutations in *RECQL4* gene (Kitao et al, 1999; Wang et al, 2002). Western blot analysis using a human RECQL4 antibody (Fig S1B) showed no detectable expression of RECQL4 in the patient fibroblasts, in contrast to fibroblasts from sex- and age-matched controls (Fig 2A) (De et al, 2012). Interestingly, expression of RECQL4 in three cancer cell lines (HeLa, HEK293, and U2OS) significantly exceeds that in fibroblasts. The cause and the functional consequences of this are yet unclear.

The Rothmund–Thomson syndrome patient's cells more frequently showed micronuclei (Fig 2B), consistent with the reported chromosome instability (Beghini et al, 2003; Miozzo et al, 1998). Analyzing mitotic spindles by indirect immunofluorescence using $\alpha$-tubulin and $\gamma$-tubulin antibodies revealed that spindles in patient cells have normal size and microtubule density, but are often tilted with respect to the substrate (Figs 2C and S2B). Spindle mis-orientation is reportedly correlated with spindle microtubule defects and chromosome misalignment observed upon down-regulation of Spindly, CLIP-170, and GTSE1 (Bendre et al, 2016; Tame et al, 2016). However, in contrast to CLIP-170 down-regulation, we did not observe an increase of "polar" chromosomes close to the spindle poles. Thus, although the mechanism of spindle tilting in the RECQL4-deficient patient cell lines might be different form that observed upon CLIP-170 down-regulation, the patient cells show mitotic defects.

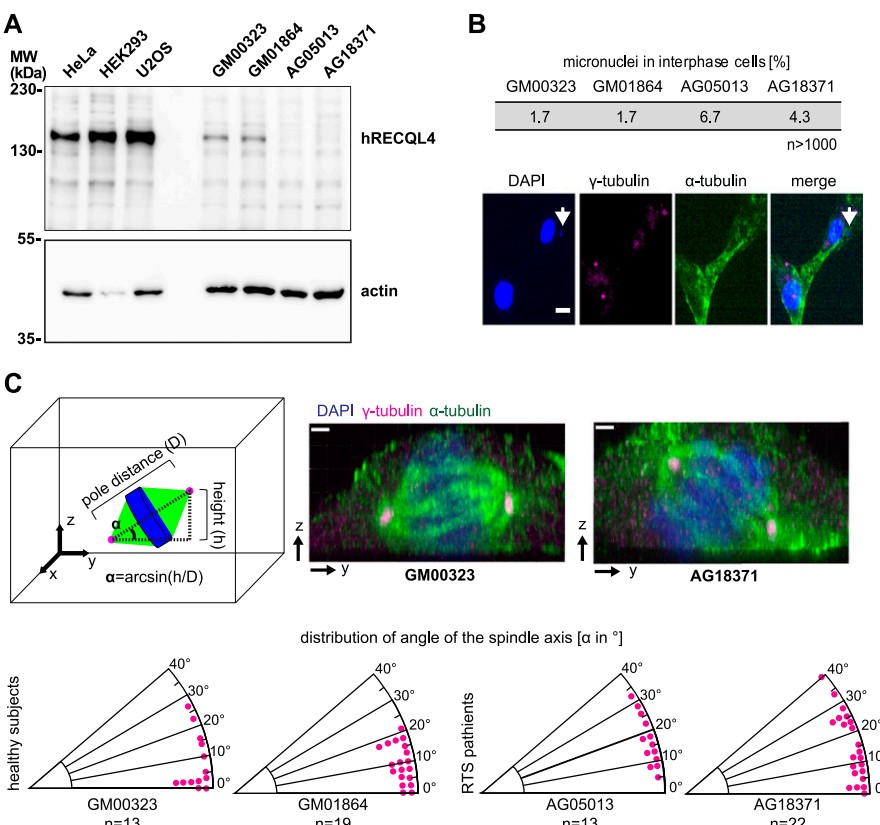

**Figure 2. Fibroblasts from Rothmund–Thomson syndrome patients have abnormal spindle axis and more micronuclei.**
**(A)** Expression of RECQL4 in HeLa, HEK293T, and U2OS immortalized cell lines is compared with the expression in healthy (GM00323, GM01864) and Rothmund–Thomson syndrome patient (AG05013, AG18371) fibroblasts, using human RECQL4 antibody. **(B)** Rothmund–Thomson syndrome fibroblasts (AG05013, AG18371) show increased amount of micronuclei as compared with healthy fibroblasts (GM00323, GM01864). More than 1,000 interphase cells per cell line were analyzed for the presence of micronuclei (DAPI stained dots) in the cytoplasm, which was identified by the α-tubulin staining. The pictures show a Rothmund–Thomson syndrome fibroblast (AG05013) with a micronucleus (arrow). Scale bars: 5 μm. **(C)** Fibroblasts were stained with α-tubulin (green) and γ-tubulin (magenta) antibodies and chromatin with DAPI (blue). The tilting of the spindle axis with respect to the culture plate was quantitated based on γ-tubulin staining (centrosomes) as in the scheme. The pictures show examples of spindle axis lateral views from a healthy (GM00323) or Rothmund–Thomson syndrome (AG18371) fibroblast. The plot shows the angle of the mitotic spindle axis with respect to the culture plate. The difference between the two control fibroblast cell lines GM00323 and GM01864 and one patient cell line (AG05013) has $P$ values of 0.02 and 0.01, respectively, the $P$ values of the control and the second patient cell line (AG18371) are 0.06 each. Scale bars: 1.5 μm.

## RECQL4 is required for maintaining mitotic chromosome alignment

The data obtained so far revealed that RECQL4 is a MAP that acts in mitosis. To analyze its role in detail, we used *Xenopus* egg extracts where spindle assembly and function can be conveniently studied (Hannak & Heald, 2006). Many Ran-regulated MAPs, because of their nuclear localization signal, are found in the nucleus in interphase and interact with the spindle in mitosis (Cavazza & Vernos, 2015). Indeed, when chromatin was added to egg extracts and then biochemically re-isolated, RECQL4 was found on interphase but not on mitotic chromatin (Fig 3A). In contrast, CAP-G, a component of the condensin complex, behaved the opposite way. Consistent with this, RECQL4 was found by immunostaining in the nucleus in interphase but on spindle microtubules in mitotic extracts (Fig 3B). Depletion of RECQL4 from egg extracts (Fig 3B) or depolymerisation of spindle microtubules by nocodazole (Fig S3A) abolished RECQL4 staining, showing the specificity of the observed nuclear and spindle labeling. Interestingly, RECQL4 was not found on the chromatin of the mitotic spindle. RECQL4 also bound to bipolar spindle structures induced in the absence of chromatin by recombinant RanGTP (Fig S3B) and localized on the spindle apparatus in tissue culture cells (Fig 3C). Thus, RECQL4 does not localize on mitotic chromatin, at least in the experimental conditions tested, but consistent with its identification as MAP, localizes to the microtubule part of the spindle. Nevertheless, depletion of RECQL4 from *Xenopus* egg extracts did not affect spindle formation in an obvious manner (Fig 3B). But, the chromatin was not stably positioned in the center of the spindle.

To analyze this in more detail, we followed the time course of spindle assembly in egg extracts. Depletion of RECQL4 did not visibly affect mitotic spindle assembly kinetics (Fig 3D). After 80 min, 74% (±2% SD, three independent experiments) of chromatin structures assembled bipolar spindles as compared with 78% (±10%) in control-depleted extracts. However, in control extracts, 81% (±7%) of the chromatin structures showed a proper chromosome alignment, whereas this level was reduced to 27% (±11%) in RECQL4-depleted extracts. During the process of spindle assembly, chromosomes were initially located in the center of the spindle in both mock and RECQL4-depleted extracts. This is expected because chromosomes drive spindle assembly through the function of Ran-GTP (Cavazza & Vernos, 2015). In RECQL4-depleted extracts, chromosomes were then scattered within the spindle with time (Fig 3D). To confirm that the observed defect was specifically caused by the lack of RECQL4, we added mRNAs corresponding to *Xenopus* or human RECQL4 to the depleted extract at the beginning of the assay, which were then translated in the egg extracts (Fig 3E). Translation of *Xenopus* RECQL4 fully rescued the chromatin alignment defect, and translation of the human protein to slightly lesser extent (Fig 3E). Together, these data show that lack of RECQL4 results in unstable chromatin alignment, consistent with the misaligned chromosome phenotype and the increased occurrence of micronuclei in tissue culture cells.

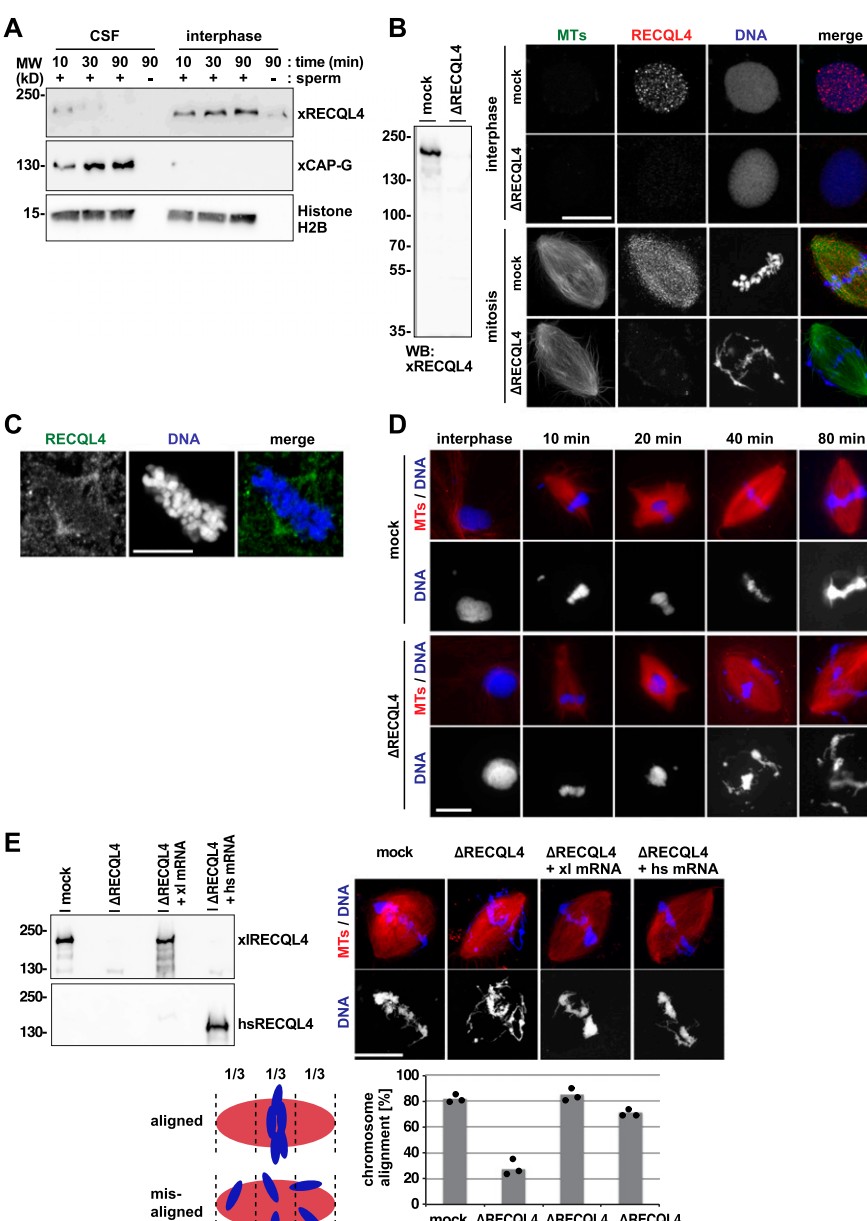

**Figure 3. RECQL4 is not necessary for spindle assembly but required for chromosome alignment.**
**(A)** RECQL4 binds to chromatin in interphase but not in mitosis. Sperm chromatin was incubated with Cytostatic factor–arrested M-phase *Xenopus* egg extract (CSF) or interphase extract prepared from the CSF extract by addition of 0.4 mM CaCl₂. At indicated time points, chromatin was isolated by centrifugation and immunoblotted for indicated proteins. Histone H2B serves as an indicator of chromatin recovery. **(B)** RECQL4 localizes in the nucleus during interphase and on spindle microtubules (MTs) during mitosis. CSF extracts were immunodepleted with control beads (mock) or RECQL4 antibody–coated beads (ΔRECQL4) and depletion efficiency was checked by Western blotting (left). These extracts were incubated with 0.4 mM CaCl₂, Alexa 488-labled tubulin (green in overlay) and sperm chromatin to allow nuclear assembly. For observing spindle assembly, the extracts were supplemented with mock or depleted CSF extract. Cy3-labeled *Xenopus* RECQL4 antibody (red in overlay) was added to the reactions 10 min before fixation. DNA was stained with DAPI. Scale bar, 20 μm. **(C)** Human IMR-90 fibroblasts were pre-extracted, fixed, and stained with an antibody against human RECQL4. DNA was counterstained with DAPI. Scale bar, 10 μm. **(D)** RECQL4 depletion causes chromosome misalignment only after spindle assembly is completed. Sperm was incubated in interphase extract supplemented with Cy3-labled tubulin and cycled back to mitosis by adding fresh CSF extract. At each time point, aliquots were fixed, stained with DAPI, and analyzed by microscopy. Scale bar, 20 μm. **(E)** Depletion of RECQL4 from extract and add-back of *Xenopus* (xl) or human (hs) RECQL4 using mRNAs. The depleted CSF extract was pre-incubated with mRNAs for 30 min, subsequently supplemented with sperm, Cy3-labled tubulin, and 0.4 mM CaCl₂, and incubated for another 90 min. The interphase extract was cycled back to mitosis by addition of CSF extract for 80 min. Samples were fixed, stained with DAPI, and analyzed by confocal microscopy. Chromosome alignment was quantified taking into account all bipolar spindle structures. Proper chromosome alignment was defined as all chromosomes located within the central third of the spindle. Columns show the average of three independent experiments and circles indicate individual data points. Extracts at the end of the assay were analyzed by Western blotting with *Xenopus* or human antibodies. Scale bar, 20 μm.

## Mitotic chromosome misalignment is independent from DNA replication and repair

As RECQL4 is reportedly involved in DNA replication (Sangrithi et al, 2005), we examined if the observed phenotype was caused by the inhibition of DNA replication. When sperm chromatin was added to cytostatic factor–arrested M-phase *Xenopus* egg extract (CSF extract), bipolar spindles assembled, despite the fact that the sperm DNA is not replicated here (Hannak & Heald, 2006) (Fig 4A). Yet, RECQL4-depleted CSF extracts assembled spindles as control extracts but still caused chromosome misalignment with time (Fig 4B). In contrast, inhibition of DNA replication by aphidicolin in extracts and recapitulation of a complete cell cycle did not induce chromosome misalignment (Fig S3C and D). RECQL4 depletion from cycling egg extracts did not abolish, but delayed, DNA replication indicated by a reduced dUTP incorporation at 45 min (Fig 4C), consistent with previous reports (Sangrithi et al, 2005; Matsuno et al, 2006). At 90 min, notably, the time point when extracts are cycled back to mitosis, DNA was eventually replicated to a similar degree in control and RECQL4-depleted extracts.

Considering the known role of RECQL4 in DNA repair (Kumata et al, 2007), we tested whether this function is linked to unstable chromosome alignment observed. DNA damage was induced in egg extract by the addition of the restriction enzyme EcoRI (Kumata et al, 2007) but did not cause chromosome misalignment (Fig S3D and E). Together, these results indicate that the chromosome misalignment observed in RECQL4-depleted extracts is independent of the protein's function in DNA replication and damage repair.

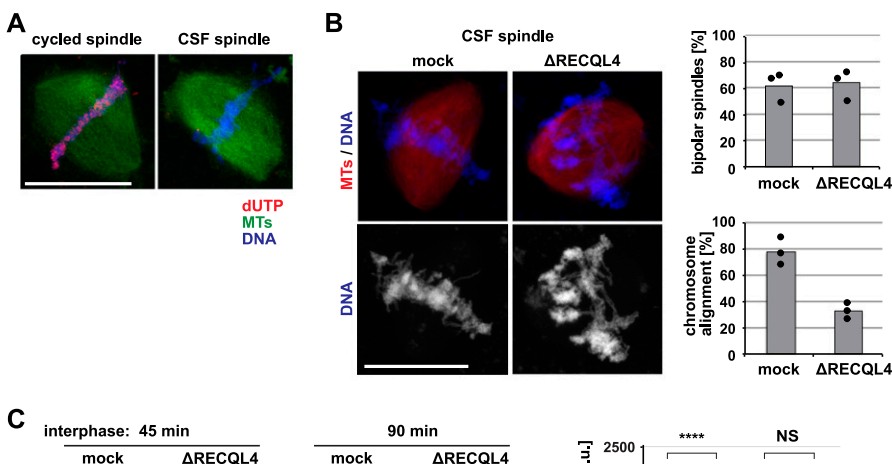

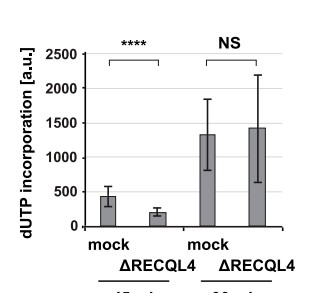

**Figure 4. Mitotic chromosome misalignment is independent of DNA replication.**
**(A)** DNA is replicated in cycled but not CSF extracts. To assemble cycled spindles, sperm was incubated in interphase extract and cycled to mitosis in the presence of Cy3-labeled dUTP. To assemble CSF spindles, sperm was incubated in CSF extract in the presence of Cy3-labeled dUTP. **(B)** Chromosome misalignment in RECQL4-depleted CSF extracts. To assemble CSF spindles, sperm was incubated in CSF extract in the presence of Cy3-labeled tubulin. Samples were fixed, stained with DAPI, and analyzed by confocal microscopy. The frequency of bipolar spindles was counted taking all chromatin structures into account. Chromosome alignment was quantified analyzing all bipolar spindle structures identified. Columns show the average of three independent experiments and circles indicate individual data points. **(C)** RECQL4 depletion delays but does not prevent DNA replication. Sperm was incubated in interphase extract in the presence of Cy3-labeled dUTP. Samples were fixed at 45 and 90 min, stained with DAPI, and analyzed by confocal microscopy. dUTP intensity on chromatin was quantified using image J. Error bars represent SD. n > 20 structures, N = 2 experiments. ****$P < 0.0001$; NS (not significant) $P > 0.05$ (*t* test, two-tailed). Scale bars, 20 $\mu$m.

## RECQL4 is required for microtubule stability and kinetochore attachment

Because RECQL4 is a MAP, we asked whether it also directly affects microtubule stability. In a first test, we added 50 ng/ml of the microtubule depolymerizing drug nocodazole to in vitro assembled spindles for additional 10 min. Whereas in control extracts spindle microtubules persisted, they depolymerized nearly completely in RECQL4-depleted extracts (Fig 5A). Importantly, chromosomes became focused, suggesting that the chromosome misalignment occurs as a consequence of dysfunctional microtubules. Spindle stability was fully rescued when depleted extracts were complemented with RECQL4 mRNA for in vitro translation, confirming that RECQL4 stabilizes spindle microtubules.

To further analyze a potential global function of RECQL4 in microtubule stabilization, we also used RanGTP (Carazo-Salas et al, 1999) and artificial chromatin beads (Heald et al, 1996) to induce bipolar spindle formation. Both in control and RECQL4-depleted extracts, microtubule structures were assembled in comparable numbers and with similar microtubule density (Figs 5B and C and S4A and B). Addition of 50 ng/ml nocodazole destabilized microtubules to a similar degree in control and RECQL4-depleted extracts. Furthermore, centrosome-induced microtubule polymerization in egg extracts was normal upon RECQL4 depletion (Fig S4C). This suggests that RECQL4 is not required for general microtubule assembly and stability.

Our data indicate that the defects associated with RECQL4 depletion are only seen in an assay that requires chromatin-, centrosomally-, and kinetochore-assembled microtubules. However, we did not detect an obvious stabilizing effect of RecQL4 on chromatin and centrosomally nucleated microtubules alone. In turn, the microtubule–kinetochore interaction reportedly stabilizes spindle microtubules against depolymerization (Emanuele & Stukenberg, 2007). We therefore speculated that the RECQL4 depletion–associated defects might be caused by a loss of, or faulty, microtubule–kinetochore interaction. To test this hypothesis, we stained spindles assembled in cycled extracts with antibodies against the kinetochore marker Ndc80/Hec1. When analyzing single slice images, the kinetochore pairs in the RECQL4-depleted extracts were often misaligned with respect to the spindle axis (Fig 6A). Surprisingly, an increased inter-kinetochore distance was observed on spindles lacking RECQL4 (Fig 6A), suggesting either abnormal tension or cohesion defects of sister chromatids. Addition of excess nocodazole (6 $\mu$g/ml) for 10 min completely depolymerized spindle microtubules (Fig 6A). Inter-kinetochore distance then decreased to a similar level in both mock and RECQL4-depleted extracts, indicating that the larger kinetochore distance in the depleted extract is due to an aberration of microtubule function.

Consistent with the above results in egg extracts, immunostaining of human fibroblasts with the kinetochore marker CREST showed significantly larger inter-kinetochore distances in metaphase cells of the Rothmund–Thomson syndrome patients (Fig 6B and C) as compared with the control. BubR1 signals in metaphase cells decreased in both control and patient cells compared with prophase (Fig 6B), indicating that microtubule-kinetochore attachment is established.

Interestingly, in the presence of 10 ng/ml nocodazole instead of 50 ng/ml, RECQL4-depletion did not result in a complete loss of microtubule mass of in vitro assembled spindles (Fig 5A and D). But, this lower nocodazole concentration rescued metaphase chromosome alignment suggesting that reduced microtubule dynamics by nocodazole complemented RECQL4 depletion. This effect was

masked at higher nocodazole concentrations due to a decrease in microtubule stability that resulted in a complete loss of microtubule production (Fig 5A).

### RECQL4 microtubule binding is critical for its function in chromatin alignment

The data presented so far establishes RECQL4 as a microtubule interacting protein involved in metaphase chromosome alignment. To see whether RECQL4–microtubule interaction is functionally connected to its role in chromosome alignment, we mapped the microtubule-binding region of *Xenopus* RECQL4 on aa 421–594 (Fig S5A–C) that includes the NLS (Woo et al, 2006; Burks et al, 2007). On several nuclear MAPs, the NLS and its neighboring regions are known as their microtubule-binding sites (Yokoyama, 2016). We generated a RECQL4 version lacking this region (Δ546–594) (Fig 7A), which was not capable of MT binding (Fig 7B) but still could interact with chromatin (Fig S5D). When depleting RECQL4 in *Xenopus* egg extracts, we observed chromosome misalignment as before (Fig 7C). This depletion phenotype was rescued by addition of the wild-type RECQL4 mRNA but not the mutant version lacking the NLS region. On the other hand, a K758M mutant, corresponding to the helicase dead K508M mutation in human RECQL4 (Rossi et al, 2010), did rescue the chromosome misalignment (Fig 7A and D). These results indicate that microtubule binding of RECQL4 and its function on the mitotic spindle is directly connected to chromosome alignment, but its function as DNA helicase is not.

Almost all mutations found in patients with the Rothmund–Thomson syndrome patients are nonsense or frameshift mutations in the middle and C-terminal region of RECQL4, whereas the N-terminal region is not affected (Larizza et al, 2010; Siitonen et al, 2009). When the mRNA encoding different C-terminal truncations of *Xenopus* RECQL4 was added back to depleted egg extracts, we observed a partial rescue of the chromosome alignment phenotype depending on the size of the protein (Fig 7E). Importantly, the aa 1–594 fragment resembling patient mutants (Kitao et al, 1999) and lacking the helicase domain rescues the misalignment some extent. Thus, truncated versions of RECQL4 might partially fulfill RECQL4 cellular functions allowing the survival of patients, whereas complete loss of RECQL4 results in embryonic lethality as confirmed in mice (Mann et al, 2005). However, it should be noted that correlation of the in vitro assays with the patient situation is hampered by the fact that mutations might affect the RNA/protein stability of RecQL4 in human cells but not in our extract experiments. For example, consistent with other Rothmund–Thomson syndrome cell lines, RECQL4 could not be detected by Western blotting in the patient fibroblast cell lines analyzed in this study (Fig 2A).

## Discussion

The function of the RECQL4 helicase has been assigned to various cellular processes (Croteau et al, 2012b) including DNA replication, DNA damage response, and telomere maintenance, similar to other RECQ family proteins (Croteau et al, 2014). Mutations in human *RECQL4* gene raise Baller-Gerold, RAPADILINO, and Rothmund–Thomson syndrome, the latter being the most extensively characterized. The Rothmund–Thomson syndrome is marked by chromosomal fragility resulting in developmental defects and cancer predisposition. Cells of patients suffering from the disease display severe chromosomal instability (Miozzo et al, 1998; Beghini et al, 2003) consistent with observations that hypomorphic RECQL4 variants in mice result in aneuploidy and cancer predisposition (Mann et al, 2005).

The prevailing hypothesis has been that cellular defects and organismic pathologies arise from losing the primary function of RECQL4's activity on DNA during its replication and/or repair in interphase. Perturbed replication and unrepaired DNA lesions could ameliorate chromosomal instability in RECQL4-deficient individuals. Here, we provide an alternative direction of thought for RECQL4-associated pathologies showing a, so far, unrecognized role of RECQL4 in mitotic spindle function.

### RECQL4 regulates chromosome alignment independently of DNA replication and damage response

Prompted by the observation that RECQL4 binds mitotic microtubules, we used a cell-free system that allowed us to dissect the function of RECQL4 in interphase from its role in mitosis. Although RECQL4 depletion from egg extracts delayed DNA replication as reported (Sangrithi et al, 2005; Matsuno et al, 2006), replication caught up to the same degree as in control extracts at the time point when the system entered mitosis (Fig 4C). It is therefore unlikely that defects in DNA replication cause chromosome misalignment in cell-free extracts depleted of RECQL4. Likewise, spindles assembled in CSF extracts, a process occurring without DNA replication, still showed chromosome misalignment upon RECQL4 depletion (Fig 4B). In contrast, neither inhibition of DNA replication nor induction of DNA damages resulted in these defects (Fig S3). Thus, chromosome misalignment caused by the absence of

---

**Figure 5. RECQL4 is required for microtubule stability.**
**(A)** RECQL4 is required for microtubule (MT) stability. Cycled spindles were assembled as in Fig 3C and D and treated with 50 ng/ml nocodazole for additional 10 min. Samples were fixed, stained with DAPI, and analyzed by confocal microscopy. MT intensity was quantified from two independent experiments with more than 30 chromatin structures per condition. ****$P < 0.0001$ ($t$ test, two-tailed). **(B)** RanGTP-induced spindles were assembled in CSF extracts and treated with or without 50 ng/ml nocodazole for an additional 10 min. Samples were fixed and analyzed. MT intensity was quantified on more than 20 spindle-like structures per condition. NS (not significant) $P > 0.05$ ($t$ test, two-tailed). **(C)** DNA-bead spindles were assembled in cycled extracts and incubated with or without 50 ng/ml nocodazole for an additional 10 min. Samples were fixed, stained with DAPI, and analyzed by confocal microscopy. MT intensity was quantified on more than 30 DNA bead clusters, each containing 15–40 beads. NS (not significant) $P > 0.05$ ($t$ test, two-tailed). **(D)** Low concentrations of nocodazole rescue the chromosome misalignment observed upon RECQL4 depletion. Cycled spindles were assembled and subsequently treated with 10 ng/ml nocodazole for additional 10 min. Samples were fixed and analyzed by confocal microscopy. Frequency of bipolar spindles was counted taking all chromatin structures into account. Chromosome alignment was quantified analyzing all bipolar spindle structures identified. Columns show the average of two independent experiments and circles indicate individual data points. Scale bars, 20 $\mu$m.

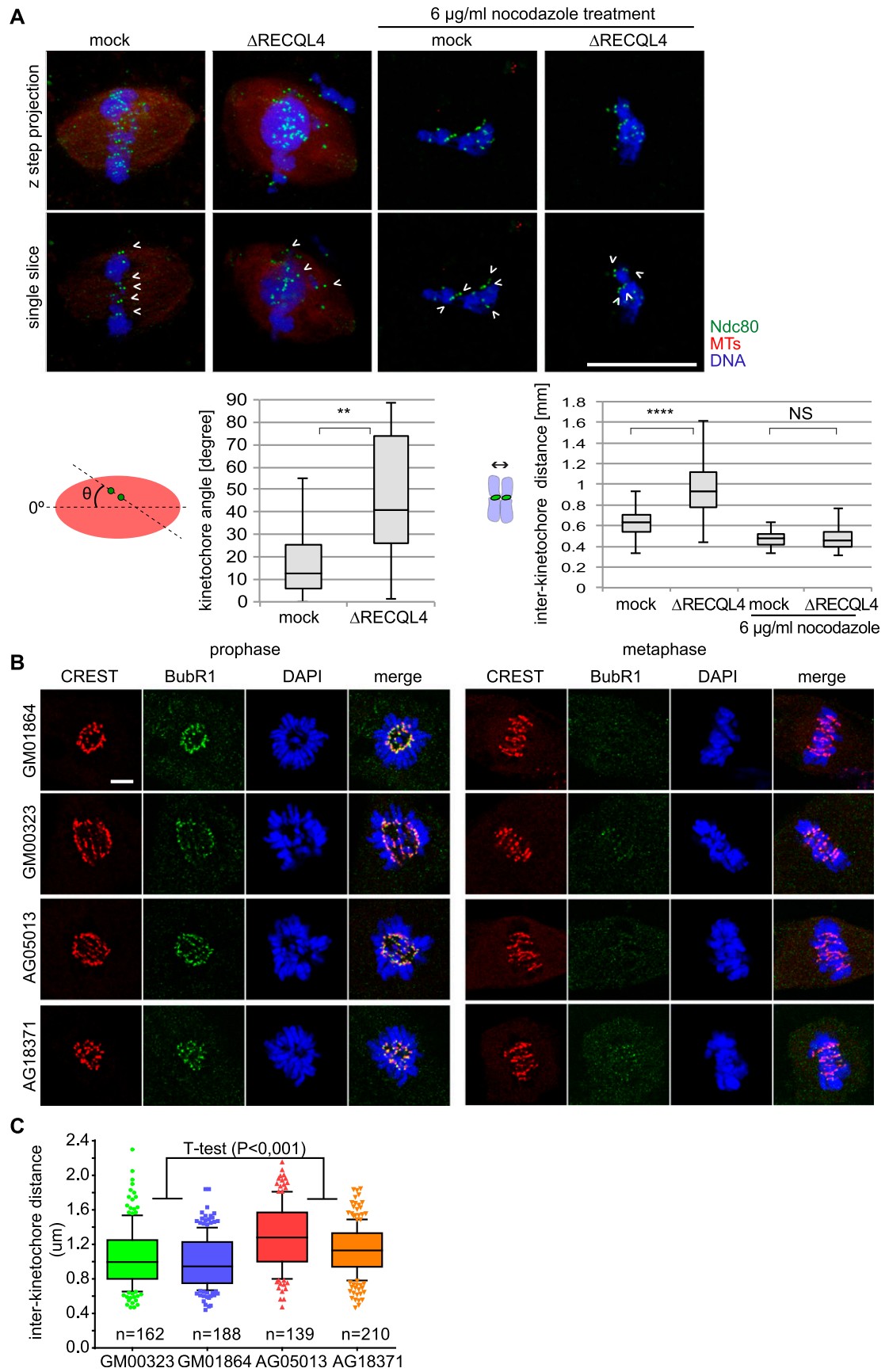

RECQL4 is not due to the defects of DNA replication and/or damage response.

Prior work suggested a role for RECQL4 in the establishment or protection of centromeric cohesion (Mann et al, 2005). However, high concentration of nocodazole treatment in RECQL4-depleted extract reduced inter-kinetochore distance to control level (Fig 6A). This suggests that chromosome misalignment observed in the depleted extract is, most likely, not due to a cohesion problem.

### RECQL4 is a mitotic MAP regulating kinetochore microtubule stability and inter-kinetochore distance

In turn, our experiments provide evidence for a novel task of RECQL4 as a regulator of spindle function, the loss of which may directly raise chromosomal instability. We find that RECQL4 binds to chromatin in interphase but then works as a microtubule-binding protein with a mitotic spindle function. Using cell-free mitotic extracts, we demonstrate that microtubule binding of RECQL4 is required for stable chromosome alignment in metaphase. Consistent with this, we observe misaligned chromosomes in HeLa cells upon RECQL4 down-regulation and micronuclei formation in fibroblasts from Rothmund–Thomson syndrome patients.

In monastrol-treated human cells, RECQL4 depletion induces asymmetric monoasters (Fig 1F and G), similar to plus-end stabilizers such as CLIP170 and CLASPs (Stolz et al, 2015). It therefore seems possible that RECQL4 functions as a plus-end microtubule stabilizer or a protein that antagonizes a plus-end microtubule de-polymerizer, similar to GTSE1, which inhibits mitotic centromere-associated kinesin (Bendre et al, 2016). Like RECQL4, GTSE1 decorates spindle microtubules, and its depletion in cells leads to chromosome misalignment and spindle mis-orientation. However, also, the microtubule bundling protein DRG1 (Schellhaus et al, 2017) or the centrosomal protein NuMA (Stolz et al, 2015) generate asymmetric monoasters in the presence of monastrol.

Interestingly, 50 ng/ml nocodazole depolymerizes in vitro assembled spindles when RecQL4 is depleted. This is seen in an assay that relies on the microtubule assembly activity of centrosomes, chromatin, and kinetochores, but not when tested with isolated chromatin/RanGTP or centrosomes (Figs 5 and S4). This may suggest that RECQL4 affects kinetochore microtubule dynamics. Indeed, the lack of RECQL4 increases the inter-kinetochore distance, both in vitro and in patient fibroblasts, again suggesting a function related to kinetochore-microtubule dynamics. A recent article suggests that RECQL4 interacts and stabilizes the aurora kinase B (Fang et al, 2018), which corrects kinetochore–microtubule attachment and ensures biorientation of sister chromatids. A better characterization of the role of RECQL4 in kinetochore–microtubule dynamics will be necessary to fully understand the mechanistic origin of the chromosome alignment defects.

### A novel function of chromosomes on their own alignment

Of the five RECQ family DNA helicases in vertebrates, only RECQL4 has been identified by our NLS-MAP purification suggesting a dual function in interphase and mitosis. Immunofluorescence shows that RECQL4 excludes from chromatin but binds to spindle microtubules in mitosis (Fig 3B). This behavior is reminiscent of the group of proteins that dissociate from mitotic chromatin and temporally regulate microtubules in mitosis (Yokoyama & Gruss, 2013). Chromosomes are the passengers segregated by the mitotic spindle to the emerging daughter cells. Yet, it has become clear that chromatin is rather an organizer of mitosis: via the small GTPase Ran and Ran-activated MAPs, chromatin initiates microtubule polymerization and organizes microtubules into a bipolar spindle early in mitosis (Cavazza & Vernos, 2015). Here, we demonstrate that microtubule binding of RECQL4 is controlled by the Ran GTPase as well. Binding of importin α and β blocks RECQL4 microtubule interaction, and the inhibition is overcome by RanGTP (Figs 1A and S1). We have previously shown that TPX2 stimulates microtubule nucleation around chromatin, once released from the inhibitory effect of importin α and β by RanGTP (Gruss et al, 2001), and CHD4 stabilizes and elongates the microtubules and is essential for spindle assembly (Yokoyama et al, 2013). In addition, the motor protein kinesin-14 bundles microtubules and is important for spindle assembly and proper spindle pole organization (Weaver et al, 2015). Independently of spindle assembly, chromatin also maintains spindle microtubules during anaphase through the function of imitation SWI (Yokoyama et al, 2009). Our findings presented here extend the regulatory role of chromosomes in mitosis. Via the Ran-activated MAP RECQL4, chromosomes control their own alignment in the metaphase plate, important for faithful chromosome segregation. Interestingly, a helicase dead RECQL4 mutant rescues the alignment defect in egg extract, supporting the idea that for this function RECQL4 does not act on chromatin but on microtubules.

In summary, we show here that the disease-mutated protein RECQL4 relocalizes from chromatin to microtubules in mitosis to contribute to mitotic microtubule regulation. Besides its function in the nucleus in interphase and in the mitochondria, RECQL4, thus, plays an important role for spindle function in chromosome alignment. The alignment is crucial for accurate chromosome segregation and cell division (Maiato et al, 2017). Chromosome alignment defects are among the multiple pathways that could lead to

**Figure 6. RECQL4 depletion or malfunction increases inter-kinetochore distance.**
**(A)** Cycled spindles are assembled as in Fig 3C and D and incubated for an additional 10 min with or without 6 μg/ml nocodazole. Samples were fixed, spun down on coverslips, stained for a kinetochore marker Ndc80 and DAPI, and analyzed by confocal microscopy. Maximum intensity projections are shown in the upper row. Single confocal slices (lower row) were used to detect kinetochore pairs (arrow heads) for further analysis. Quantitation shows the inter-kinetochore distance (right) and the relative angles of sister kinetochore pairs (left), measured with respect to the spindle pole to pole axis. n > 30 kinetochore pairs from > 6 structures. Note that after RECQL4 depletion, sister kinetochore pairs do not align to the pole to pole axis. Scale bar, 20 μm. ****P < 0.0001; **P < 0.01; NS (not significant) P > 0.05 (t test, two-tailed). **(B)** Immunofluorescence staining of control (GM00323, GM01864) and Rothmund–Thomson syndrome patient (AG05013, AG18371) fibroblasts with the kinetochore marker CREST and checkpoint marker BubR1. Scale bar, 5 μm. **(C)** Inter-kinetochore distance was measured in metaphase cells of control (GM00323, GM01864) and Rothmund–Thomson syndrome patient (AG05013, AG18371) fibroblasts based on CREST signals for the kinetochore pairs attached to microtubules (identified by the absence of the BubR1 signal) after 3D reconstruction. (n) indicates the number of kinetochore pairs measured per fibroblast line. P < 0.001 (t test, two-tailed).

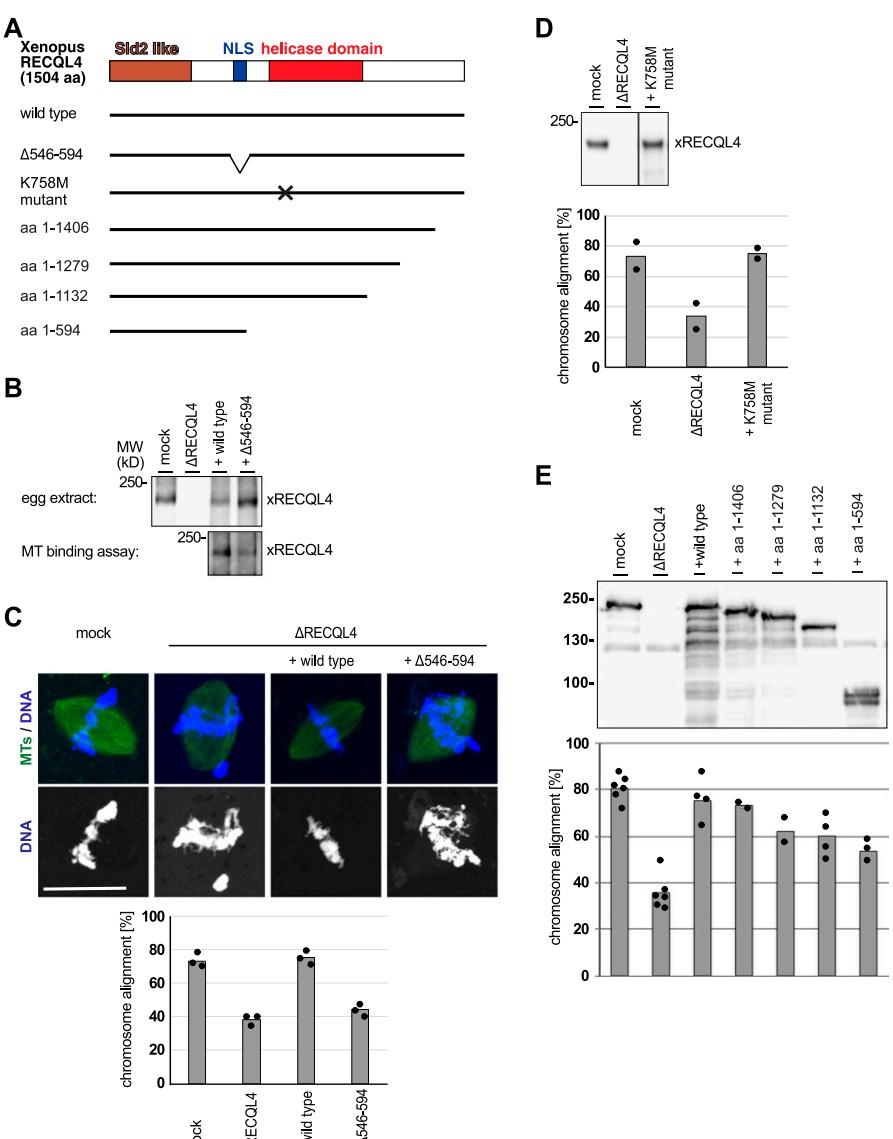

**Figure 7. The microtubule-binding region of RECQL4 is required for chromosome alignment.**
**(A)** Schematic representation of *Xenopus* RECQL4 and constructs used in add-back reactions. The position of the N-terminal Sld2-like domain, involved in DNA replication, the NLS, and the helicase domain are indicated. The Δ546–594 mutant lacks the NLS region. The K758M point mutant is a known helicase-defective RECQL4 version (Rossi et al, 2010). **(B)** RECQL4-depleted (ΔRECQL4) CSF extract was incubated with wild-type or Δ546–594 mRNAs for 90 min. Expression of the recombinant proteins were confirmed by Western blotting using *Xenopus* antibodies. The resulting extracts were used for microtubule (MT) binding assay. **(C)** Sperm was incubated in control (mock) or RECQL4-depleted (ΔRECQL4) CSF extracts supplemented with the indicated mRNAs for cycled spindle assembly in the presence of Alexa 488-labled tubulin. At the end of the reaction, samples were fixed and stained with DAPI for microscopy. Chromosome alignment was quantified analyzing all bipolar spindle structures identified. Columns show the average of three independent experiments and circles indicate individual data points. Scale bar, 20 μm. **(D)** Cycled spindles assembled as in (C) but with the helicase-defective K758M mutant. Depletion and add-back efficiency was analyzed at the end of the assay by Western blotting. Chromosome alignment was quantified. Columns show the average of two independent experiments and circles indicate individual data points. **(E)** Cycled spindles assembled as in (C) but supplemented with mRNA encoding for wild-type or different C-terminal RECQL4 truncations. Depletion and add-back efficiency was analyzed at the end of the assay by Western blotting. Chromosome alignment was also quantified. Columns show the average of at least two independent experiments and circles indicate individual data points.

chromosome instability (Thompson et al, 2010; Gordon et al, 2012), a hallmark of cancer cells. The novel function of RECQL4 described here thus provides additional molecular insights to understand patient symptoms in Rothmund–Thomson as a consequence of chromosome instability.

## Materials and Methods

### Recombinant proteins and antibodies

A cDNA clone (IRAKp961N20331Q) covering the complete *Xenopus* RECQL4 cDNA was subcloned into pFastBac HTa (Invitrogen). The protein was expressed in Sf21 insect cells, and purified on TALON beads (BD Biosciences), dialyzed to CSF-XB buffer (10 mM K-Hepes, 100 mM KCl, 3 mM MgCl$_2$, 0.1 mM CaCl$_2$, 50 mM sucrose, and 5 mM EGTA, pH 7.7) containing 10% glycerol and 1 mM DTT, and used for microtubule sedimentation assay. For antibody production against *Xenopus* RECQL4, the protein was also expressed in insect cells but solubilized from inclusion bodies with 6 M guanidine hydrochloride. The protein was purified on TALON beads, dialyzed to 8 M urea, and used for immunization in rabbits. Human RECQL4 cDNA (GI: 284005308) was in vitro synthesized (GenScript). An N-terminal fragment of human RECQL4 (aa 1–831) was cloned into a pET28a vector (Novagen). The corresponding protein was expressed in BL21 (DE3) *E. coli*, purified on Ni-NTA-Agarose (QIAGEN), and used for antibody production in rabbits. To determine the microtubule-binding region of RECQL4 in sedimentation assays, *Xenopus* RECQL4 fragments were subcloned into a pET28a vector, expressed in BL21 (DE3) cells, purified with Ni-NTA-Agarose, and dialyzed to 20 mM Tris, 300 mM NaCl, pH 8.0. Importin α, importin β, and RanQ69L-GTP were expressed in *E. coli* and purified with TALON beads (Yokoyama et al, 2014).

For in vitro translation in egg extracts, *Xenopus laevis* and human RECQL4 were cloned by PCR into a pCS2+ vector. The RECQL4 Δ546–594 mutant was created by replacing the original *Xenopus* sequence between the Bsu36I and EcoRV by an in vitro synthesized gene fragment (Integrated DNA Technologies, Leuven, Belgium). The DNA replacement does not change the coding aa sequence but creates BsrGI and BglII sites. These sites are used to replace the intermediate sequence by an in vitro synthesized gene fragment encoding for the Δ546–594 mutant.

The following published and commercial antibodies were used: XCAP-G for Western blot at 1 µg/ml (Magalska et al, 2014), chTOG antibody for Western blot at 1 µg/ml (Yokoyama et al, 2014), *Xenopus* Ndc80/Hec1 antibody at 1 µg/ml for immunofluorescence (Emanuele & Stukenberg, 2007), human histone H2B antibody (Millipore, used 1:1,000 for Western blotting), BubR1 (Millipore, at 1:500 for immunofluorescence), CREST antibody (Antibody Inc., at 1:200 for immunofluorescence), phospho-histone H2AX (Cell Signaling, at 1:200 for immunofluorescence), α-tubulin (mouse DM1A; Sigma-Aldrich, at 1:200 for immunofluorescence), and γ-tubulin at 1 µg/ml for immunofluorescence (Barenz et al, 2013). Secondary antibodies for immunofluorescence were Alexa-Fluor-488-anti-mouse, Alexa-Fluor-647-anti-human, and Alexa-Fluor-647-anti-rabbit (from Life Technologies, used at 1:1,000).

## Microtubule binding assays

0.1 µM recombinant RECQL4 was incubated with 2 µM taxol-stabilized microtubules for 15 min, and centrifuged at 20,000 *g* in a TLA120.2 rotor (Beckman) for 10 min at RT. The supernatant and pellet was analyzed by Coomassie staining or Western blot. The assay was also performed in the presence or absence of recombinant 2 µM importin α, 2 µM importin β, and 5 µM RanQ69L-GTP, a dominant positive mutant of Ran locked in the GTP-bound state.

HeLa nuclear extract (4C Biotech) was diluted to 1 mg/ml with CSF-XB buffer. *Xenopus* CSF egg extracts were diluted 1:3 with CSF-XB buffer to a concentration of about 30 mg/ml. After centrifugation with TLA-100.2 rotor at 100,000 *g* for 10 min at 4°C, the supernatant was incubated at RT in the presence or absence of 2 µM taxol-stabilized microtubules, 2 µM importin α, 2 µM importin β, and 5 µM RanGTP for 15 min. The samples were centrifuged at 100,000 *g* for 10 min at 20°C, and the pellets were incubated with CSF-XB supplemented with 500 mM NaCl for 5 min and centrifuged again. The supernatant (eluate) was analyzed by Western blotting.

## *Xenopus* egg extracts and cell-free assays

Cytostatic factor–arrested M-phase *Xenopus laevis* egg extracts (CSF extracts) were prepared as described (Hannak & Heald, 2006). In short, *Xenopus* eggs were dejellied by cysteine treatment, washed with XB buffer (10 mM K-Hepes, 100 mM KCl, 1 mM MgCl$_2$, 0.1 mM CaCl$_2$, and 50 mM sucrose, pH 7.7) and subsequently CSF-XB buffer, and crushed by centrifugation at 20,000 *g* for 20 min in a SW55 Ti rotor (Beckman) at 16°C. The straw-colored middle layer was recovered as a CSF extract. Endogenous RECQL4 was depleted from CSF extracts by two rounds of incubation with 60% (vol/vol) Protein A Dynabeads (Invitrogen) coupled with *Xenopus* RECQL4 antibodies.

For spindle assembly in cycled extract, CSF extract was supplemented with demembranated sperm (Eisenhardt et al, 2014) and

Cy3 or Alexa488-labeled tubulin, and driven into interphase by addition of 0.4 mM CaCl$_2$ and incubation at 20°C for 90 min. Samples were cycled to mitosis by addition of fresh CSF extract and incubating at 20°C for 80 min. For CSF spindle assembly, CSF extracts were incubated with demembranated sperm and Cy3-labeled tubulin at 20°C for 80 min. Microtubule density around sperm or beads was quantified using Matlab (The MathWorks). To examine RECQL4 localization, Cy3-labled *Xenopus* RECQL4 antibody was added to the assembly reactions at 5 ng/ml and incubated for additional 10 min before fixation. For rescue experiments, RECQL4 mRNA (mMESSAGE mMachine kit; Life Technologies) was added at 100 ng/µl at the beginning of the reactions. Cycled spindles were treated with indicated concentrations of nocodazole for the last 10 min and, for measurement of inter-kinetochore distance, stained for a kinetochore marker Ndc80/Hec1. Single confocal slices were used to find kinetochore pairs. The inter-kinetochore distance and the relative angles of sister kinetochore pairs against the spindle pole to pole axis were measured using Image J.

DNA replication was monitored by incorporation of 5 µM Cy3-labeled dUTP and inhibited, where indicated, by 50 µg/ml aphidicolin (Matsuno et al, 2006). DNA damage was induced by addition of 0.2 units/µl EcoRI restriction enzyme (Kumata et al, 2007), and monitored by immunostaining for phospho-Histone H2AX.

Chromatin re-isolation from CSF or interphase extract, DNA-bead spindle assembly, RanGTP-induced microtubule/spindle assembly, and centrosomal microtubule assembly were performed as described previously (Yokoyama et al, 2014).

## Cell culture and transfection

Human healthy (GM00323 and GM01864) and Rothmund–Thomson syndrome (AG05013 and AG18371) fibroblasts (Coriell Institute) were maintained in MEM supplemented with 2 mM L-glutamine, 15% FBS, and 500 units/ml penicillin–streptomycin (all from Gibco). All HeLa cell lines were cultured in DMEM supplemented with 2 mM L-glutamine, 10% FBS, and 500 units/ml penicillin–streptomycin (all from Gibco). For the HeLa H2B–mCherry and EGFP-α-tubulin cell line (a kind gift from Daniel Gerlich), the same medium was additionally supplemented with 0.5 µg/ml puromycin (Gibco) and 500 µg/ml G-418 (Geneticin; Life Technologies) as described (Held et al, 2010). The siRNA knockdown experiments were performed with the following siRNA oligonucleotides against RECQL4: siRECQL4#1 (s17991), 5′-GGCUCAACAUGAAGCAGAAtt-3′, siRECQL4#2 (s17993), 5′-CCCAAUACAGCUUACCGUAtt-3′, siRECQL4#3 (HSS190281), 5′-GAUGUCACAGUGAGGuCCCAGAUUU-3′, (from Life Technologies). siRNA AllStar (from QIAGEN) was used as negative control. 40 nM from each siRNA were used for transfecting HeLa cell suspensions with Lipofectamine RNAiMAX (Invitrogen) according to the manufacturer's instructions.

For immunofluorescence analysis, fibroblasts were grown for 48 h on eight-well µ-slide chambers (Ibidi) and fixed with 4% PFA. After 1 h in blocking buffer (PBS + 0.1% Triton-X100 + 3% BSA), the samples were incubated for 2 h with α-tubulin and γ-tubulin antibodies in blocking buffer at RT. As secondary antibodies Alexa-Fluor-647-anti-Rabbit and Alexa-Fluor-488-anti-mouse (Life Technologies) were used 1 h at RT. 1 µg/ml DAPI was added for 10 min and the samples were mounted with a medium optimized for

fluorescence microscopy in μ-slides (Ibidi). All cells in meta-phase present in two wells of an Ibidi chamber per fibroblast cell line were analyzed on a LSM780 confocal with a 63 × 1.4 NA objective (z > 50 slices per cell: z-step 233 nm, pinhole 20 μm). The tilting of the spindle axes with respect to horizontal plane was determined as described previously (Toyoshima & Nishida, 2007) using IMARIS (Bitplane). In brief, the γ-tubulin staining marking the spindle poles of cells in metaphase was used to determine distances between the two spindle poles in XY and in Z. Then, the spindle angle to the substratum was calculated using inverse trigonometry.

For micronuclei analysis, the total well surface of the samples were imaged on a LSM5live using the 20× Air 0.8 NA objective in a tile scan mode covering all the cells with 30 × 33 tiles and 75 Z-slices. Maximum intensity projections in Z from the tile files were generated in ZEN (Zeiss). The resulting files were analyzed in IMARIS, determining the percentage of cells with visible micronuclei covering the diagonal from the upper right corner to the center of the tile field until more than 1,000 total interphase cells were considered.

For analysis of inter-kinetochore distances, fibroblasts were seeded on glass coverslides in 24-well plates (Greiner Bio-One), fixed after 24 h with 4% PFA, immunostained with centromere (CREST) and BubR1 antibodies, and mounted with mowiol 4-88 (Calbiochem). Acquisition from seven to 10 random metaphase cells per cell line was performed as z-Stacks (z-scaling 255 nm/Pinhole 20 μm) with a confocal Zeiss LSM780 equipped with a Plan-Apochromat 63×/1.4 Oil DIC M27 objective and 405 nm-DPSS, 488 nm-Argon, and 633 nm-Diode Lasers. IMARIS (Bitplane) was used for measuring the inter-kinetochore distance within sister kinetochores identified by the CREST signal in those kinetochores clearly devoid of BubR1 signal and attached to microtubules.

For the quantitation of lagging chromosomes, chromatin bridges, and ultra-fine bridges, HeLa cells, stably expressing H2B–mCherry, were seeded on glass coverslides and transfected with siRNA oligonucleotides. After 48 h, cells were fixed with 4% PFA and stained with an anti-ERCC6L/PICH antibody (Abnova #H00054821-Do1p). Full 3D volumes of more than 20 random late anaphase cells from different replicates were imaged on a Zeiss LSM710 confocal microscope equipped with a Plan-Apochromat 63×/1.4 Oil objective and 488 nm and 561 nm lasers using a pinhole of 1 AU. Lagging chromosomes and regular chromatin bridges were visualized and counted based on the H2B-mCherry signal, ultra-fine chromatin bridges based on the Plk1-interacting checkpoint helicase (PICH) staining using ZEN software.

### Live-cell imaging experiments

HeLa cells expressing H2B-mCherry and EGFP-α-tubulin were transfected with siRNA oligonucleotides in eight-well μ-slide chambers (Ibidi) and, after 24 h, were imaged for 48 h in a LSM 5 live confocal microscope (Zeiss) equipped with a heating and $CO_2$ incubation system (Ibidi). Seven 3.6-μm-spaced optical z-sections at various positions every 3 min were acquired with a Plan-Apochromat 20× NA 0.8 objective and a 488-nm and 561-nm diode lasers controlled by ZEN software. For the analysis, maximum intensity projections in Z were generated in ZEN for every position and converted into temporal image sequences with the free licensed AxioVision software (LE64; V4.9.1.0). Afterward, segmentation, annotation, classification, tracking of cells during mitosis, and extraction of galleries with the identified cell tracks were performed using the Cecog Analyzer (http://www.cellcognition.org/software/cecoganalyzer) (Held et al, 2010) The percentage of tracks with persistently misaligned meta-phase chromosomes was identified as clearly isolated chromosomes separated from the metaphase plate in two or more consecutive frames, and visually determined. Microsoft Excel and GraphPad Prism were used for data analysis from more than 100 cell tracks per condition in three independent experiments.

### Evaluation of monoastrol mitotic spindles

HeLa cells, seeded on glass coverslides, were transfected with siRNA oligonucleotides in 24-well plates (Greiner Bio-One). After 72 h, the cells were incubated with 70 μM monastrol (Sigma-Aldrich) and with or without 2 nM taxol (Stolz et al, 2015; Schellhaus et al, 2017). Samples were fixed with 4% PFA, stained with anti-human centromere (CREST) and α-tubulin antibodies and DAPI, and mounted with mowiol 4-88 (Calbiochen). The imaging from five to eight random positions per siRNA and condition was performed as z-stacks (z-scaling 350 nm/Pinhole 25 μm) with a confocal Zeiss LSM780 equipped with a Plan-Apochromat 40×/1.3 Oil DIC M27 objective and 405 nm-DPSS, 488 nm-Argon, and 633 nm-Diode Lasers.

### Immunostaining of spindles

Detection of RECQL4 on human spindles was performed in IMR90 cells after pre-extraction in 0.3% TX100 in PHEM buffer for 2 min and fixation in pre-warmed PFA for 10 min at RT. Fixed samples were incubated with an antibody against human RECQL4 for 3 h at RT and further visualized with an anti-rabbit Cy3 secondary antibody. Imaging was performed on a Zeiss LSM880 confocal system using a Plan-APOCHROMAT 63×/1.4 Oil objective. Images of optimized confocal stacks (Zeiss ZEN software) in the respective channels were used to generate maximum projections in Image J 64 1.45S.

### Whole-cell extracts of tissue culture cells

For comparing RECQL4 expression in U2OS, HEK293, HeLa, and human healthy (GM00323 and GM01864) and Rothmund–Thomson syndrome (AG05013 and AG18371) fibroblasts, 120,000 cells from each cell line were collected by centrifugation at 100 g for 1 min, washed once with PBS, centrifuged at 15,700 g for 2 min, resuspended in 60 μl loading buffer (200 mM Tris, pH6.8, 1,000 mM sucrose, 10% SDS, 0.1% bro-mophenol blue + 1/10 β-mercaptoethanol), boiled for 5 min, and analyzed by Western blotting.

For assessing RECQL4 knock-down efficiency in siRNA experiments, HeLa cells expressing H2B-mCherry and EGFP-α-tubulin cells were transfected with siRNA oligonucleotides in eight-well μ-slide chambers (Ibidi). 48 and 72 h post-transfection, the cells were washed three times in the wells with PBS and directly taken up in 50 μl loading buffer, boiled for 5 min, and analyzed by Western blotting.

### Statistical analysis

Microsoft Excel and GraphPad Prism were used for statistical analysis. The data were tested for normality by D'Agostino & Pearson omnibus normality test when possible. Then, variances were compared by F test ($P < 0.05$). Two-tailed t test was performed

if a Gaussian distribution for the data series could be assumed and they had no significantly different variances. Two-tailed *t* test with Welch's correction was performed if Gaussian distribution could be assumed for the data series but they had significantly different variances. Mann–Whitney test was performed if a Gaussian distribution could not be assumed (*P*-value legend ****0.0001 < *P*; ***0.001 < *P*; ** 0.01 < *P*; * 0.05 < *P*).

## Supplementary Information

## Acknowledgements

We thank T Stukenberg for *Xenopus* Ndc80 antibody and D Gerlich for the HeLa H2B–mCherry and EGFP-*α*-tubulin cell line. We also thank the Light Microscope Facility of Max Planck Institute for Developmental Biology, Tuebingen, for imaging and Animal Facility in European Molecular Biology Laboratory for *Xenopus* RECQL4 antibody production. This study was supported by JSPS KAKENHI grant no. JP16K21749 to H Yokoyama and by European Research Council grant (309528 CHROMDECON) and by the German Research Foundation (DFG, AN377/3-2 and AN377/6-1) to W Antonin.

### Author Contributions

H Yokoyama: conceptualization, formal analysis, supervision, funding acquisition, validation, investigation, visualization, and writing—original draft, review, and editing.
D Moreno-Andres: conceptualization, formal analysis, investigation, visualization, and writing–review and editing.
SA Astrinidis: investigation.
Y Hao: formal analysis and investigation.
M Weberruss: formal analysis and investigation.
AK Schellhaus: investigation.
H Lue: investigation.
Y Haramoto: investigation.
OJ Gruss: supervision and writing–review and editing.
W Antonin: conceptualization, funding acquisition, supervision, and writing–original draft, review, and editing.

### Conflict of Interest Statement

The authors declare that they have no conflict of interest.

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
