## [Reviewer comments · Life Science Alliance]

Life Science Alliance

Chromosome alignment maintenance requires the MAP RECQL4, mutated in the Rothmund-Thomson syndrome

Wolfram Antonin, Hideki Yokoyama, Daniel Moreno, Susanne Astrinidis, Yuqing Hao, Marion Weberruss, Hongqi Lue, and Oliver Gruss

DOI: <https://doi.org/10.26508/lsa.201800120>

Corresponding author(s): Wolfram Antonin, RWTH University and Hideki Yokoyama, ID Pharma / National Institute of Advanced Industrial Science and Technology

Review Timeline:

Submission Date:	2018-07-01
Editorial Decision:	2018-07-26
Revision Received:	2018-12-17
Editorial Decision:	2019-01-18
Revision Received:	2019-01-25
Accepted:	2019-01-25

Scientific Editor: Andrea Leibfried

Transaction Report:

July 26, 2018

Re: Life Science Alliance manuscript #LSA-2018-00120-T

Prof. Wolfram Antonin
RWTH University
Institute for Biochemistry and Molecular Cell Biology
Pauwelsstraße 30
Aachen 52074
Germany

Dear Dr. Antonin,

Thank you for submitting your manuscript entitled "Chromosome alignment maintenance requires the MAP RECQL4, mutated in the Rothmund-Thomson syndrome" to Life Science Alliance. The manuscript was assessed by expert reviewers, whose comments are appended to this letter. We invite you to submit a revision if you can address the reviewers' key concerns.

As you will see, the reviewers appreciate your data. However, they also note some discrepancies and think that additional support for your conclusions is needed. All three reviewers provide constructive input on how address the current discrepancies and to provide additional support. Additionally, they suggest to strengthen the scope of your work to warrant the title and discussion about a potential link to Rothmund-Thomson syndrome. We would thus like to invite you to submit a revised version addressing the concerns raised by the reviewers, which seem all straightforward to address following their suggestions. We think it would be good to also include disease-relevant RECQL4 mutants in your analyses as outlined by the reviewers.

-- High-resolution figure, supplementary figure and video files uploaded as individual files: See our detailed guidelines for preparing your production-ready images, <http://life-science-alliance.org/authorguide>

B. MANUSCRIPT ORGANIZATION AND FORMATTING:

Full guidelines are available on our Instructions for Authors page, <http://life-science-alliance.org/authorguide>

Thank you for this interesting contribution to Life Science Alliance. We are looking forward to receiving your revised manuscript.

Sincerely,

Reviewer #1 (Comments to the Authors (Required)):

This manuscript reports a previously unreported function for the RECQL4 helicase (RECQ) as a microtubule binding protein that regulates chromosome alignment. The evidence in support of this role includes microtubule pelleting assays (including regulation of microtubule binding by Ran), cytological analysis of the progression of mitosis in control and RECQ depleted extracts and some localization data. Much of the analysis is performed in *Xenopus* extracts that allows the authors to avoid a large number of indirect effects. Overall the evidence that RECQ regulates mitotic functions and can associate with microtubules is convincing. However, there are a number of ways that this manuscript would need to be improved to render it suitable for publication in LSA.

The introduction focuses extensively on the human disease implications of mutations in RECQ, "Yet, a correlation between genotype and phenotype in different pathological RECQL4 mutant alleles is complex and uncertain" and raises the possibility that the disease associated mutations in this gene maybe due to the mitotic abnormalities due to loss of function of RECQ. However, there is no evidence provided that addresses this important issue. This seems like a missed opportunity because the assay shown in fig 7 would readily lend itself to this analysis. Either the authors should provide an overview of the known mutations and the diseases they are associated with and a series of assays of mutation in the rescue assay or they should dramatically de-emphasize this portion of the manuscript (with a strong recommendation for the former). This analysis should include a helicase defective variant

The authors state, "We have recently identified RECQL4 as a potential mitosis specific microtubule associated protein (MAP) [20]" RECQ does not appear in reference 20 in the main text or the supplemental tables. This is a strange omission that requires clarification.

The authors show a modest accumulation of RECQ on spindles in *Xenopus* extracts. It would be helpful to show the localization of RECQ in mitotic cells in a cultured cell line, to better assess the degree to which RECQ partitions onto the spindle.

The functional data are somewhat contradictory:

- a) Fig 1FG, RECQ depletion results in asymmetric spindles as is seen in extracts depleted of plus tip stabilizers.
- b) Fig 5 A, spindle microtubules in RECQ depleted extracts are sensitive to microtubule destabilizing agents, consistent with a model in which RECQ stabilizes microtubules.
- c) bead spindles and asters lacking RECQ are not more sensitive to microtubule depolymerizing agents, suggesting RECQ is specifically acting on kinetochores, not microtubules in general.
- d) the effects of depletion of RECQ can be rescued by low dose nocodazole, consistent with RECQ acting to destabilize microtubules.

These phenotypes are difficult to rationalize into a consistent model and the authors do not address this apparent confusion. In particular, RECQ depletion only affects spindles when kinetochores are present. This is not consistent with a general microtubule binding protein. In addition, it is not clear why *all* the microtubules in a spindle would be nocodazole sensitive when RECQ is depleted, when asters are not.

Other issues:

1- Is the $\Delta 546-594$ mutant dominant? There seems to be a chromosome decondensation/condensation phenotype caused by the mutant protein in the rescue assay (fig 7C), that is not observed in the simple depletion.

3- Both DNA and microtubules are negatively charged polymers. The mapped domain could also regulate chromatin binding.

4- The data in figure 1 F,G is consistent with, but does not prove that, RECQ is a plus end microtubule stabilizer. It is not clear that this assay adds much to the manuscript.

5- The experiments in figure 4 C,D should include a control lacking nocodazole but depleted of RECQ.

Reviewer #2 (Comments to the Authors (Required)):

This interesting paper provides evidence supporting a new function for the helicase RECQL4 mutated in Rothmund-Thomson syndrome. The authors first show that RECQL4 binds to microtubules in a ranGTP-regulated manner, and that its depletion or mutation in human cells causes chromosome mis-alignment during mitosis. Interestingly, they find that RECQL4 depletion from cell-free *Xenopus* extracts causes chromosome mis-alignment only after spindle assembly has been completed, and provide evidence that RECQL4 promotes microtubule stability. The authors also show that the mitotic role of RECQL4 is independent, at least in the *Xenopus* model, of known functions in DNA replication and repair.

This paper provides insight into the functions of a poorly studied protein mutated in human diseases, with a compelling and technically sound series of experimental results to validate each conclusion. I have no major technical concerns bar the points below.

Specific comments.

The authors have nicely leveraged the power of the cell-free *Xenopus* system to separate RECQL4 functions in mitosis from those in DNA repair and replication. Although the authors' results using human cells confirm that RECQL4 depletion or inactivation can cause mitotic anomalies, the possibility that repair/replication defects may also contribute in this setting is not yet convincingly excluded. The paper's main conclusion would be strengthened if further evidence could be provided. For example, does a nuclear-localized form of the $\Delta 421-594$ RECQL4 mutant support DNA replication/repair functions but not chromosome alignment in human cells? (I recognize that since the 421-594 region includes the endogenous NLS, reincorporation of an alternative NLS may be required.)

On the same point, the authors' conclusions would also be strengthened if they could localize RECQL4 to mitotic structures in human cells besides *Xenopus* (Fig 3B). (I appreciate that their anti-RECQL4 1-871 antibody recognizes a non-specific band in Westerns, so fluorophore tagging may be necessary.) Is localization affected by disease-associated mutations?

RECQL4 encodes a helicase activity that is often deleted or inactivated by human disease-associated mutations. Does this activity play a role during mitosis?

The authors correctly point out in the abstract and text that the reason why different RECQL4 mutations cause distinct human phenotypes remains unclear. But I think it is premature to claim that the current results provide "an intriguing molecular explanation for the disease-associated phenotypes of Rothmund-Thomson syndrome" until they can demonstrate more clearly that different mutations are analysed against the mitotic function to elicit stronger genotype-phenotype

correlations. For example, it is intriguing that C-terminal deletions in murine RECQL4 cause aneuploidy, although that region of RECQL4 is not (yet) implicated in mitosis. Short of addressing this issue experimentally, which may take a significant effort, changes to the text to better explain and tone down these points should be made.

p.8 - "despite of its reported function as a DNA helicase, RECQL4 does not localize on mitotic chromatin". Is such a result under these experimental conditions expected for a helicase implicated in early stages of DNA replication, and in DSB repair? A more stringent test (eg., replicating or damaged chromatin) may be required. If so, this statement (and later parts of the discussion) should be modified.

Reviewer #3 (Comments to the Authors (Required)):

In this manuscript Yokoyama and colleagues study the potential mitotic role of the RECQL4 helicase in human cells and *Xenopus laevis* extracts. The authors propose that RECQL4 regulates the interaction between kinetochores and microtubules, and that as such it is required for efficient chromosome alignment in mitosis. The main findings of the study are that 1) RECQL4 binds to microtubules and is specifically associated to the mitotic spindle; 2) that it is required for chromosome alignment and efficient mitotic progression; 3) that its depletion results in abnormal kinetochore-microtubule interactions; 4) and that these functions are not linked to the known roles of RECQL4 in DNA replication.

Overall, these findings are novel, interesting and pertinent to the field. However, at present stage several of the proposed findings are not well supported by the presented data, and the manuscript would highly benefit from major revision experiments.

Specifically:

1) Microtubule-binding: while the in vitro MT-binding experiments are of good quality, the immunofluorescence experiments should be improved, as at the moment there is only one immunofluorescence image of RECQL4 on a bipolar spindle in mitotically arrested frog extracts. First, the authors should show that the localization around the chromatin disappears if the extracts are treated with a microtubule-depolymerizing drug. Second, do the authors also see RECQL4 on the mitotic spindle in human cells, and does this localization depend on microtubules? Given that the authors claim a specific role for RECQL4 on kinetochore-microtubules, it would be easier to test in human cells if RECQL4 is enriched on kinetochore-microtubules. As more minor point the authors could also measure the affinity of RECQL4 for microtubules in their in vitro assay. This would provide more substantial evidence, but is not essential for the paper.

2) Effects on chromosome alignment: The presented experiments consistently show that loss of RECQL4 results in chromosome alignment both in human cells and in frog extracts. This is a strong point of the study. The authors also show that different siRNAs delay mitosis to varying degrees.

This variability is however, not really explained. Does it depend on the siRNA efficiency (note that the experiments were done after 24 hours depletion, but the western blot measuring the depletion levels were taken after 48 hours? At minimum, the authors should comment on this. Also do the cells align all their chromosomes before anaphase onset, or do the unaligned chromosomes persist at anaphase onset. This might help to document how alignment defects can lead to chromosome mis-segregation.

3) The authors show that depletion of RECQL4 results in spindle microtubules that are more easily depolymerized by nocodazole and that inter-kinetochores are higher when RECQL4 is not present. Based on these data the authors suggest a defect in kinetochore-microtubule interactions. The first issue is that if the kinetochore-microtubule interaction is weaker, as suggested by the lower kinetochore-microtubule stability, then the inter-kinetochore distances should be lower, not higher. In the same line of thought, a weaker kinetochore-microtubule interaction should lead to metaphase plates with a few BubR1 positive kinetochores that are not fully attached. The second issue is that the authors measure inter-kinetochore distances in single image slices, essentially in 2D, which can lead to measurements artifacts. It is essential that the authors study these aspects more carefully, before drawing broad conclusions, as at the moment their results paradoxically suggest a weaker kinetochore-microtubule interaction that leads to stronger pulling forces on sister-kinetochores. First, does RECQL4 depletion change the dynamics of kinetochore-microtubules? Instead of using nocodazole treatment, which can lead to a high variability, it might be helpful to use cell lines expressing photo-activatable GFP-tubulin to record the tubulin turnover in kinetochore-microtubules (Zhai et al.; 1995), this would give a much more reliable measure. Second, the authors should use 3D measurements to measure inter-kinetochore distances; also the authors should not use CREST staining in human cells, as CREST gives a diffuse staining all along the centromeric DNA, due to the fact that it recognizes several proteins at centromeres. Antibodies against CENP-A or Ndc80 are a much better choice. Finally, could it be that RECQL4 affects centromeric cohesion (e.g. CAPD2 or CAPD3 binding to chromatin). This would explain the higher-inter-kinetochore distances, and might explain the reduced microtubule stability, as kinetochore-based tension would fail to stabilize microtubules. Exploring this aspect might help to sort out this paradox.

4) Spindle orientation defect: In figure 2B the authors report a spindle orientation defect. First is the difference in spindle angle significant between the different cell lines? Second, the authors point to a recent study of the Medema laboratory that showed that chromosome alignment defects can lead to spindle orientation defects. However, not every unaligned chromosome leads to spindle orientation defects, but rather specific "polar" chromosomes that are behind or close to a spindle pole (Tame et al., 2016). The unaligned chromosomes in this study, however, appear to be between the spindle poles and the metaphase plate, and they are not thought to lead to spindle orientation defects. Or do the authors see a correlation between the spindle orientation defect and unaligned chromosomes?

5) Asymmetric spindle: the authors report in Figure 1F that RECQL4 depletion leads to "asymmetric" spindles in monastrol-treated cells. It is not clear how the authors can decide how a monopolar spindle is "asymmetric" or not, at least not based on the images provided by the authors. This should be either much better explained, or removed, as it is not an essential point of the story.

Reviewer #1

This manuscript reports a previously unreported function for the RECQL4 helicase (RECQ) as a microtubule binding protein that regulates chromosome alignment. The evidence in support of this role includes microtubule pelleting assays (including regulation of microtubule binding by Ran), cytological analysis of the progression of mitosis in control and RECQ depleted extracts and some localization data. Much of the analysis is performed in *Xenopus* extracts that allows the authors to avoid a large number of indirect effects. Overall the evidence that RECQ regulates mitotic functions and can associate with microtubules is convincing. However, there are a number of ways that this manuscript would need to be improved to render it suitable for publication in LSA.

The introduction focuses extensively on the human disease implications of mutations in RECQ, "Yet, a correlation between genotype and phenotype in different pathological RECQL4 mutant alleles is complex and uncertain" and raises the possibility that the disease associated mutations in this gene maybe due to the mitotic abnormalities due to loss of function of RECQ. However, there is no evidence provided that addresses this important issue. This seems like a missed opportunity because the assay shown in fig 7 would readily lend itself to this analysis. Either the authors should provide an overview of the known mutations and the diseases they are associated with and a series of assays of mutation in the rescue assay or they should dramatically de-emphasize this portion of the manuscript (with a strong recommendation for the former). This analysis should include a helicase defective variant.

Our answer: We have indeed used the complementation assay in *Xenopus* egg extracts (Figure 7) to test C-terminal truncations of RECQL4 as many of the reported RECQL4 mutations are predicted to generate a C-terminal truncated protein. Indeed, the longer the truncation, the more severe the effect on chromosome alignment (data added as Figure 7E). However, we do not know how mutations affect the stability of RecQL4 in human cells, which makes the correlation to the pathologically relevant mutations difficult. For example, RECQL4 cannot be readily detected by Western blotting in the patients' fibroblast cell lines analyzed in this manuscript. Many disease-linked mutations lie within the helicase domain and affect protein function rather than stability. As requested, we therefore tested a helicase defective variant exploiting the complementation assay in *Xenopus* egg extracts (K758M, Figure 7D). Interestingly, this mutant is able to substitute the endogenous protein for its function in chromosome alignment. We discuss these points in our revised manuscript and de-emphasize the disease aspect in the introduction.

The authors state, "We have recently identified RECQL4 as a potential mitosis specific microtubule associated protein (MAP) [20]" RECQ does not appear in reference 20 in the main text or the supplemental tables. This is a strange omission that requires clarification.

Our answer: Indeed, we have not shown the complete list of identified proteins in the mentioned publication – thanks for pointing at this. We changed the sentences in the new version of manuscript to be clear on this point.

The authors show a modest accumulation of RECQ on spindles in *Xenopus* extracts. It would be helpful to show the localization of RECQ in mitotic cells in a cultured cell line, to better assess the degree to which RECQ partitions onto the spindle.

Our answer: Although we could not visualize RecQL4 in mitotic HeLa cells to compare with corresponding knock-down samples, we were able to show spindle localization of endogenous human RECQL4 by immunofluorescence on mitotic human fibroblasts (IMR90 cells). These data have been now included into Figure S3c.

The functional data are somewhat contradictory:

- a) Fig 1FG, RECQ depletion results in asymmetric spindles as is seen in extracts depleted of plus tip stabilizers.
- b) Fig 5 A, spindle microtubules in RECQ depleted extracts are sensitive to microtubule destabilizing agents, consistent with a model in which RECQ stabilizes microtubules.
- c) bead spindles and asters lacking RECQ are not more sensitive to microtubule depolymerizing agents, suggesting RECQ is specifically acting on kinetochores, not microtubules in general.
- d) the effects of depletion of RECQ can be rescued by low dose nocodazole, consistent with RECQ acting to destabilize microtubules.

These phenotypes are difficult to rationalize into a consistent model and the authors do not address this apparent confusion. In particular, RECQ depletion only affects spindles when kinetochores are present. This is not consistent with a general microtubule binding protein. In addition, it is not clear why **all** the microtubules in a spindle would be nocodazole sensitive when RECQ is depleted, when asters are not.

Our answer: As now better explained in the manuscript, symmetric spindles are not only seen upon depletion of plus end stabilizers but also for NuMA (Stolz et al., 2015) and DRG1, which bundles microtubules (Schellhaus et al 2017). Our data using 50 ng/ml nocodazole indicates that RECQL4 gets rate limiting for general microtubule stability when tested in an assay that reflects mitotic spindle assembly. In this assay, chromatin (driven mostly by RanGTP), kinetochores and centrosomes contribute to microtubule production and stability. In contrast, chromatin beads and RanGTP only reflect the chromatin contribution. It has been previously reported that the depletion of microtubule associated proteins can affect microtubules in these assays differently (e.g. Tpx2, Xnf7) indicating that stable microtubule production in the different assays displays different sensitivities towards depletion of stabilizing MAPs. We still consider the assay employing sperm nuclei and generating a spindle from centrosomes, chromatin and kinetochores as the most relevant one, which is closest to the situation in somatic cells.

We also show a second experiment using 10 ng/ml nocodazole. Interestingly, here, the depletion of RECQL4 did not result in a further loss of microtubule mass as observed with 50 ng/ml nocodazole. It rather completely rescued metaphase chromosome alignment suggesting that, here, reduced microtubule dynamics complemented RECQL4 depletion – an effect that was masked at higher nocodazole concentrations due to a decrease in microtubule stability that resulted in a complete loss of microtubule production.

Other issues:

1- Is the $\Delta 546-594$ mutant dominant? There seems to be a chromosome decondensation/condensation phenotype caused by the mutant protein in the rescue assay (fig 7C), that is not observed in the simple depletion.

Our answer: By systematically checking other spindles in the same experiment, we did not see a chromosome decondensation/condensation-defect phenotype. We therefore replaced the image in Fig7C showing the addback of the $\Delta 546-594$ mutant to a more representative one.

3- Both DNA and microtubules are negatively charged polymers. The mapped domain could also regulate chromatin binding.

Our answer: Please note that the $\Delta 546-594$ protein also binds chromatin as presented in the chromatin re-isolation experiment (Supplementary Figure S5D). We have modified the respective sentence in the results part.

4- The data in figure 1 F,G is consistent with, but does not prove that, RECQ is a plus end microtubule stabilizer. It is not clear that this assay adds much to the manuscript.

Our answer: As indicated above, asymmetric spindles are not only seen upon depletion of plus end stabilizers but also upon NuMA and DRG1 depletion. The assay indicates that spindle dynamics is affected by RECQL4 depletion consistent with its suggested mitotic function on microtubules.

5- The experiments in figure 4 C,D should include a control lacking nocodazole but depleted of RECQ.

Our answer: We assume this points refers instead to Figure 5C,D. We have done DNA-bead spindle assembly in the absence or presence of nocodazole, and presented the data in the new figure 5C. The experiments showing RecQL4 depletion in the absence of nocodazole using sperm DNA as chromatin template and cycling extracts are shown in Figure 3D.

Reviewer #2

This interesting paper provides evidence supporting a new function for the helicase RECQL4 mutated in Rothmund-Thomson syndrome. The authors first show that RECQL4 binds to microtubules in a ranGTP-regulated manner, and that its depletion or mutation in human cells causes chromosome mis-alignment during mitosis. Interestingly, they find that RECQL4 depletion from cell-free *Xenopus* extracts causes chromosome mis-alignment only after spindle assembly has been completed, and provide evidence that RECQL4 promotes

microtubule stability. The authors also show that the mitotic role of RECQL4 is independent, at least in the *Xenopus* model, of known functions in DNA replication and repair.

This paper provides insight into the functions of a poorly studied protein mutated in human diseases, with a compelling and technically sound series of experimental results to validate each conclusion. I have no major technical concerns bar the points below.

Specific comments.

The authors have nicely leveraged the power of the cell-free *Xenopus* system to separate RECQL4 functions in mitosis from those in DNA repair and replication. Although the authors' results using human cells confirm that RECQL4 depletion or inactivation can cause mitotic anomalies, the possibility that repair/replication defects may also contribute in this setting is not yet convincingly excluded. The paper's main conclusion would be strengthened if further evidence could be provided. For example, does a nuclear-localized form of the $\Delta 421-594$ RECQL4 mutant support DNA replication/repair functions but not chromosome alignment in human cells? (I recognize that since the 421-594 region includes the endogenous NLS, reincorporation of an alternative NLS may be required.)

Our answer: We have tried to address this question in *Xenopus* egg extracts. The extracts need to be incubated after RECQL4 depletion for 60 min with the respective mRNA for an addback to resynthesize the RecQL4 versions. Unfortunately, even mock depleted extracts were incapable of DNA replication after this incubation time. In cells, the experiments are likewise difficult to perform: RECQL4 with an extra-NLS needs to be transformed and stably expressed in cells lacking endogenous RecQL4 at physiological levels. Although an interesting experiment, this is not doable in the time frame of this revision.

On the same point, the authors' conclusions would also be strengthened if they could localize RECQL4 to mitotic structures in human cells besides *Xenopus* (Fig 3B). (I appreciate that their anti-RECQL4 1-871 antibody recognizes a non-specific band in Westerns, so fluorophore tagging may be necessary.) Is localization affected by disease-associated mutations?

Our answer: We have added an experiment showing RECQL4 immunofluorescence on mitotic cells to visualize spindle localization. These data have been now included into Figure S3C as also requested by reviewer 1. As also elaborated in the reply to reviewer 1 the correlation to the human mutations is hampered by the fact that we do not know how human mutations affect the stability of RECQL4. For example, in the fibroblast cell lines analyzed in this manuscript no RECQL4 is detected by western blotting.

RECQL4 encodes a helicase activity that is often deleted or inactivated by human disease-associated mutations. Does this activity play a role during mitosis?

Our answer: As requested, the *Xenopus* version of a previously described helicase defective variant (Rossi et al. 2010) is now included in the analysis (K758M, Figure 7D). Interestingly, this mutant is able to substitute the endogenous protein for its

function in chromosome alignment suggesting that the helicase activity does not play a role during mitosis.

The authors correctly point out in the abstract and text that the reason why different RECQL4 mutations cause distinct human phenotypes remains unclear. But I think it is premature to claim that the current results provide "an intriguing molecular explanation for the disease-associated phenotypes of Rothmund-Thomson syndrome" until they can demonstrate more clearly that different mutations are analyzed against the mitotic function to elicit stronger genotype-phenotype correlations. For example, it is intriguing that C-terminal deletions in murine RECQL4 cause aneuploidy, although that region of RECQL4 is not (yet) implicated in mitosis. Short of addressing this issue experimentally, which may take a significant effort, changes to the text to better explain and tone down these points should be made.

Our answer: As suggested, we examined the C-terminal deletions of RECQL4 and found that they are not able to fully rescue the chromosome alignment phenotype (Figure 7E). Also, we carefully rephrased the text to avoid overstatements.

p.8 - "despite of its reported function as a DNA helicase, RECQL4 does not localize on mitotic chromatin". Is such a result under these experimental conditions expected for a helicase implicated in early stages of DNA replication, and in DSB repair? A more stringent test (e.g., replicating or damaged chromatin) may be required. If so, this statement (and later parts of the discussion) should be modified.

Our answer: Indeed, a number of DNA replication factors dissociate from chromatin during mitosis (e.g. Mcm2 and polymerase δ , Gillespie et al. 2007). Therefore, the dissociation of RECQL4 is not unexpected but we indeed did not check upon DNA damage indication in mitosis. We have rephrased the manuscript to be more careful on this point

Reviewer #3

In this manuscript Yokoyama and colleagues study the potential mitotic role of the RECQL4 helicase in human cells and *Xenopus laevis* extracts. The authors propose that RECQL4 regulates the interaction between kinetochores and microtubules, and that as such it is required for efficient chromosome alignment in mitosis. The main findings of the study are that 1) RECQL4 binds to microtubules and is specifically associated to the mitotic spindle; 2) that it is required for chromosome alignment and efficient mitotic progression; 3) that its depletion results in abnormal kinetochore-microtubule interactions; 4) and that these functions are not linked to the known roles of RECQL4 in DNA replication.

Overall, these findings are novel, interesting and pertinent to the field. However, at present stage several of the proposed findings are not well supported by the presented data, and the manuscript would highly benefit from major revision experiments.

Specifically:

1) Microtubule-binding: while the in vitro MT-binding experiments are of good quality, the immunofluorescence experiments should be improved, as at the moment there is only one

immunofluorescence image of RECQL4 on a bipolar spindle in mitotically arrested frog extracts. First, the authors should show that the localization around the chromatin disappears if the extracts are treated with a microtubule-depolymerizing drug. Second, do the authors also see RECQL4 on the mitotic spindle in human cells, and does this localization depend on microtubules? Given that the authors claim a specific role for RECQL4 on kinetochore-microtubules, it would be easier to test in human cells if RECQL4 is enriched on kinetochore-microtubules. As more minor point the authors could also measure the affinity of RECQL4 for microtubules in their in vitro assay. This would provide more substantial evidence, but is not essential for the paper.

Our answer: Please note that the immunofluorescence labeling is abolished on in vitro assembled spindles in RECQL4 depleted egg extracts showing the specificity of the staining (Fig 3B). We included now, as requested, an experiment which shows that RECQL4 spindle localization disappears if microtubules are depolymerized by nocodazole (Fig S3A). As also requested by the other reviewers we have added an experiment showing RECQL4 immunofluorescence on mitotic cells to visualize spindle localization in human somatic cells. These data have been now included into Figure S3C. A quantitation of RECQL4 binding to microtubules is now provided in the manuscript. This allows a comparison to other MAPs, e.g. CHD4, which binds under these conditions also microtubules (Yokoyama et al., 2013).

2) Effects on chromosome alignment: The presented experiments consistently show that loss of RECQL4 results in chromosome alignment both in human cells and in frog extracts. This is a strong point of the study. The authors also show that different siRNAs delay mitosis to varying degrees. This variability is however, not really explained. Does it depend on the siRNA efficiency (note that the experiments were done after 24 hours depletion, but the western blot measuring the depletion levels were taken after 48 hours? At minimum, the authors should comment on this. Also do the cells align all their chromosomes before anaphase onset, or do the unaligned chromosomes persist at anaphase onset. This might help to document how alignment defects can lead to chromosome mis-segregation.

Our answer: The data presented in Figure 1C, D and E are obtained from cells upon life cell imaging 24 till 72h post-transfection. Although after 48h and 72h the depletion level as judged by western blotting is quite efficient, we cannot exclude small variations. We have added according comments to the manuscript. Increased numbers of lagging chromosomes after anaphase onset were not detected after RECQL4 depletion. We now also checked for anaphase bridges and ultra-fine chromatin bridges using PICH as marker and did not observe a significant increase upon RECQL4 depletion (data now included in the manuscript).

3) The authors show that depletion of RECQL4 results in spindle microtubules that are more easily depolymerized by nocodazole and that inter-kinetochores are higher when RECQL4 is not present. Based on these data the authors suggest a defect in kinetochore-microtubule interactions. The first issue is that if the kinetochore-microtubule interaction is weaker, as suggested by the lower kinetochore-microtubule stability, then the inter-kinetochore distances should be lower, not higher. In the same line of thought, a weaker kinetochore-microtubule interaction should lead to metaphase plates with a few BubR1 positive kinetochores that are not fully attached.

Our answer: We have now discussed this apparent contradiction more carefully

The second issue is that the authors measure inter-kinetochore distances in single image slices, essentially in 2D, which can lead to measurements artifacts.

Our answer: The measurements in cells were done using 3D reconstruction using the IMARIS tool set. For the egg extract assays, the stack size of 0.5 μm is too large to allow for 3D reconstruction but they nevertheless agree with the analysis in cells.

It is essential that the authors study these aspects more carefully, before drawing broad conclusions, as at the moment their results paradoxically suggest a weaker kinetochore-microtubule interaction that leads to stronger pulling forces on sister-kinetochores. First, does RECQL4 depletion change the dynamics of kinetochore-microtubules? Instead of using nocodazole treatment, which can lead to a high variability, it might be helpful to use cell lines expressing photo-activatable GFP-tubulin to record the tubulin turnover in kinetochore-microtubules (Zhai et al.; 1995), this would give a much more reliable measure.

Our answer: We currently do not have cell lines, constructs and microscopic setting available to use photoactivatable GFP-tubulin. Thus, performing the experiment is unrealistic in the time frame of a revision. However, we assayed in the course of this study EB3-EGFP comet dynamics during metaphase in and did not find a significant difference between control and RECQL4 depleted cells with regard to comet speed (data not shown).

Second, the authors should use 3D measurements to measure inter-kinetochore distances; also the authors should not use CREST staining in human cells, as CREST gives a diffuse staining all along the centromeric DNA, due to the fact that it recognizes several proteins at centromeres. Antibodies against CENP-A or Ndc80 are a much better choice.

Our answer: The measurements in cells were done using 3D reconstruction using the IMARIS tool set. We precisely used CREST because it allows assignment of the kinetochore pairs. This marker has been used by other studies in the field (e.g. Heit et al., 2009; Mahale et al, 2016; Lagirand-Cantaloube et al., 2017).

Finally, could it be that RECQL4 affects centromeric cohesion (e.g. CAPD2 or CAPD3 binding to chromatin). This would explain the higher-inter-kinetochore distances, and might explain the reduced microtubule stability, as kinetochore-based tension would fail to stabilize microtubules. Exploring this aspect might help to sort out this paradox.

Our answer: High concentration of nocodazole treatment in RecQL4-depleted extracts reduced inter-kinetochore distance to control levels (Fig. 6A). This suggests that the chromosome misalignment observed in depleted extracts is not due to a cohesion problem. We mention this point in the discussion.

4) Spindle orientation defect: In figure 2B the authors report a spindle orientation defect. First is the difference in spindle angle significant between the different cell lines? Second, the authors point to a recent study of the Medema laboratory that showed that chromosome alignment defects can lead to spindle orientation defects. However, not every unaligned chromosome leads to spindle orientation defects, but rather specific "polar" chromosomes that are behind or close to a spindle pole (Tame et al., 2016). The unaligned chromosomes in this study, however, appear to be between the spindle poles and the metaphase plate, and they are not thought to lead to spindle orientation defects. Or do the authors see a correlation between the spindle orientation defect and unaligned chromosomes?

Our answer: The difference between the two control fibroblast cell lines and one patient cell line (AG05013) has P values of 0.02 and 0.01, the P values of the controls and the second patient cell line (AG18371) are 0.06 each. We have added this information to the figure legend. Indeed, we did not see increased numbers of polar chromosomes and commented this in the manuscript.

5) Asymmetric spindle: the authors report in Figure 1F that RECQL4 depletion leads to "asymmetric" spindles in monastrol-treated cells. It is not clear how the authors can decide how a monopolar spindle is "asymmetric" or not, at least not based on the images provided by the authors. This should be either much better explained, or removed, as it is not an essential point of the story.

Our answer: We followed the protocol described in the original paper (Stolz et al., 2015), which established the assay. As requested, we explained the procedure more precisely in the manuscript.

January 18, 2019

RE: Life Science Alliance Manuscript #LSA-2018-00120-TR

Prof. Wolfram Antonin
RWTH University
Institute for Biochemistry and Molecular Cell Biology
Pauwelsstraße 30
Aachen 52074
Germany

Dear Dr. Antonin,

Thank you for submitting your revised manuscript entitled "Chromosome alignment maintenance requires the MAP RECQL4, mutated in the Rothmund-Thomson syndrome". As you will see, while supportive of publication, the reviewers point out that a few issues still need to get addressed. We would thus like to invite you to submit a final version of your manuscript, addressing the points raised by the reviewers. Additionally, please address the following editorial points:

- please make sure that you add statistical test used and p-values obtained to each figure legend
- please note that figure panel 6C is not mentioned in the text nor legend, please fix
- figure panel 7A is not mentioned in the text, please do
- please correct figure legend S3: the last 'D' should be an 'F'
- please link your profile in our submission system to your ORCID ID (required for all corresponding authors), you and the other corresponding author should have received an email with instructions on how to do so

A. FINAL FILES:

-- High-resolution figure, supplementary figure and video files uploaded as individual files: See our detailed guidelines for preparing your production-ready images, <http://life-science-alliance.org/authorguide>

-- Summary blurb (enter in submission system): A short text summarizing in a single sentence the study (max. 200 characters including spaces). This text is used in conjunction with the titles of

papers, hence should be informative and complementary to the title. It should describe the context and significance of the findings for a general readership; it should be written in the present tense and refer to the work in the third person. Author names should not be mentioned.

B. MANUSCRIPT ORGANIZATION AND FORMATTING:

Full guidelines are available on our Instructions for Authors page, <http://life-science-alliance.org/authorguide>

Sincerely,

Andrea Leibfried, PhD
Executive Editor
Life Science Alliance
Meyershofstr. 1
69117 Heidelberg, Germany
t +49 6221 8891 502
e a.leibfried@life-science-alliance.org
www.life-science-alliance.org

Reviewer #1 (Comments to the Authors (Required)):

The authors have suitably revised the manuscript and it is essentially ready for publication.
minor points

- 1 - The data from S3C (localization of RECQL4 in mitotic cells) should be in the main figures.
- 2 - figure 6A is annotated with 6 mg/ml nocodazole, not 6 μ g/ml
- 3 - The concluding sentence of the abstract is a bit too strong. This could be an alternative cause of RT syndrome, or the syndrome could result from a spectrum of molecular defects. Perhaps "defects in mitotic chromosome alignment might be a contributing factor for RT syndrome." would be a better conclusion.

Reviewer #3 (Comments to the Authors (Required)):

This revised manuscript address the role of RECQL4 in the mitotic spindle and its contribution to chromosome segregation in the context of the Rothmund-Thomson syndrome.

While the authors have improved the manuscript and addressed several concerns of the reviewers, I still feel that the authors have missed an occasion to better characterize the effects of RECQL4 depletion on spindle microtubule dynamics. The authors have not really provided a solid direct and quantitative assessment of the contribution of RECQL4, as the only data is the fact that RECQL4-depleted spindle are more prone to microtubule depolymerization in nocodazole-treated cells. The issue is that this is rather a crude and descriptive assay, that does not provide lot of mechanistic insight. This should not preclude publication, but this concern reduces the enthusiasm for the manuscript. As a suggestion, it might be useful for the authors to note in the discussion that a better characterization of the role of RECQL4 in kinetochore-microtubule dynamics will be necessary to fully understand the mechanistic origin of the chromosome alignment defects.

January 25, 2019

RE: Life Science Alliance Manuscript #LSA-2018-00120-TRR

Prof. Wolfram Antonin
RWTH University
Institute for Biochemistry and Molecular Cell Biology
Pauwelsstraße 30
Aachen 52074
Germany

Dear Dr. Antonin,

Thank you for submitting your Research Article entitled "Chromosome alignment maintenance requires the MAP RECQL4, mutated in the Rothmund-Thomson syndrome". I appreciate the introduced changes and it is a pleasure to let you know that your manuscript is now accepted for publication in Life Science Alliance. Congratulations on this interesting work.

DISTRIBUTION OF MATERIALS:

Again, congratulations on a very nice paper. I hope you found the review process to be constructive and are pleased with how the manuscript was handled editorially. We look forward to future exciting submissions from your lab.

Sincerely,

Andrea Leibfried, PhD
Executive Editor
Life Science Alliance
Meyerohofstr. 1
69117 Heidelberg, Germany
t +49 6221 8891 502
e a.leibfried@life-science-alliance.org
www.life-science-alliance.org